# Major proliferation of transposable elements shaped the genome of the soybean rust pathogen *Phakopsora pachyrhizi*

With >7000 species the order of rust fungi has a disproportionately large impact on agriculture, horticulture, forestry and foreign ecosystems. The infectious spores are typically dikaryotic, a feature unique to fungi in which two haploid nuclei reside in the same cell. A key example is *Phakopsora pachyrhizi*, the causal agent of Asian soybean rust disease, one of the world's most economically damaging agricultural diseases. Despite *P. pachyrhizi*'s impact, the exceptional size and complexity of its genome prevented generation of an accurate genome assembly. Here, we sequence three independent *P. pachyrhizi* genomes and uncover a genome up to 1.25 Gb comprising two haplotypes with a transposable element (TE) content of ~93%. We study the incursion and dominant impact of these TEs on the genome and show how they have a key impact on various processes such as host range adaptation, stress responses and genetic plasticity.

Rust fungi are an order of >7000 species of highly specialized plant pathogens with a disproportionately large impact on agriculture, horticulture, forestry, and foreign ecosystems[1]. The infectious spores are typically dikaryotic, a feature unique to fungi in which two haploid nuclei reside in the same cell. Asian soybean rust caused by the obligate biotrophic fungus *Phakopsora pachyrhizi*, is a prime example of the damage that can be caused by rust fungi. It is a critical challenge for food security and one of the most damaging plant pathogens of this century (Fig. 1a)[2]. The disease is ubiquitously present in the soybean growing areas of Latin America, where 210 million metric tons of soybean are projected to be produced in 2022/23 (https://apps.fas.usda.gov/psdonline/app/index.html), and on average representing a gross production value of U.S. $ 115 billion per season (https://www.ers.usda.gov/data-products/season-average-price-forecasts.aspx). A low incidence of this devastating disease (0.05%) can already affect yields and, if not managed properly, yield losses are reported of up to 80%[3,4]. Chemical control in Brazil to manage the disease started in the 2002/03 growing season[4]. In the following season, ~20 million hectares of soybeans were sprayed with fungicides to control this disease (Fig. 1a)[4,5]. The cost of managing *P. pachyrhizi* exceeds $2 billion USD per season in Brazil alone.

The pathogen is highly adaptive and individually deployed resistance genes have been rapidly overcome when respective cultivars have been released[6,7]. Similarly, the fungal tolerance to the main classes of site-specific fungicides is increasing, making chemical control less effective[8–10]. Another remarkable feature for an obligate biotrophic pathogen is its wide host range, encompassing 153 species of legumes within 54 genera to date[11–13]. Epidemiologically, this is relevant as it allows the pathogen to maintain itself in the absence of soybean on other legume hosts, such as overwintering on the invasive weed Kudzu in the United States[14]. Despite the importance of the pathogen, not much was known about its genetic makeup as the large genome size (an estimated 1 Gbp), coupled to a high repeat content, high levels of heterozygosity and the dikaryotic nature of the infectious urediospores of the fungus have hampered whole genome assembly efforts[15].

In this work, we provide reference quality assemblies and genome annotations of three *P. pachyrhizi* isolates. We uncover a genome with a total assembly size of up to 1.25 Gb. Approximately, 93% of the genome consists of TEs, of which two superfamilies make up 80% of the TE content. The three *P. pachyrhizi* isolates collected from South America represent a single clonal lineage with high levels of heterozygosity. Studying the TEs in detail, we demonstrate that the expansion of TEs within the genome happened over the last 10 My and accelerated over the last 3 My, and did so in several bursts. Although TEs are tightly controlled during sporulation and appressoria

✉e-mail: Peter.vanesse@tsl.ac.uk

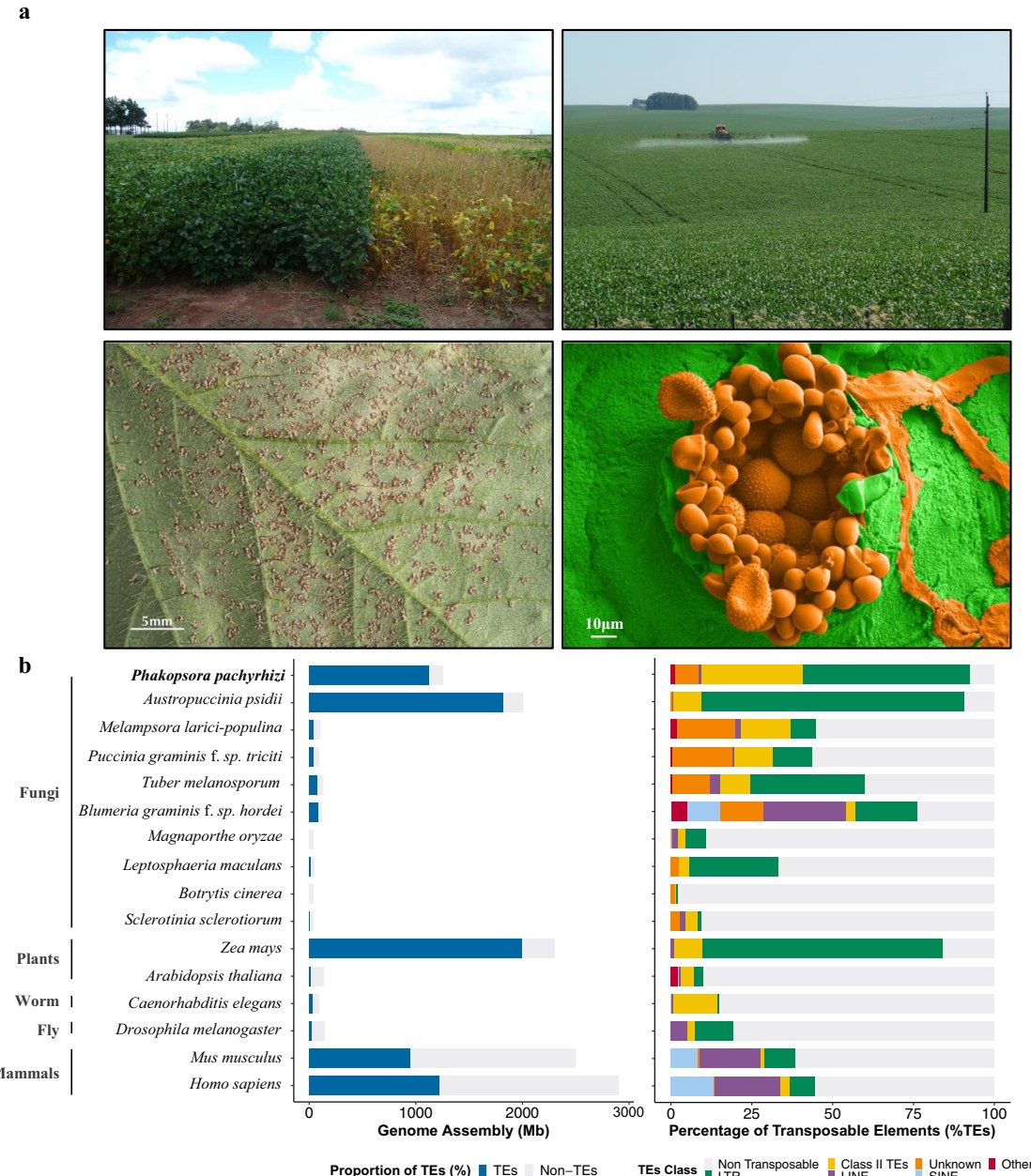

**Fig. 1 | Impact of *P. pachyrhizi* incidence in a soybean field, comparative genome assembly size, and TE content. a** Soybean field sprayed with fungicide (left) and unsprayed (right) in Brazil (top left). Soybean field being sprayed with fungicide (top right). Soybean leaf with a high level of *P. pachyrhizi* urediospores, Tan lesions (bottom left). Electron micrograph of *P. pachyrhizi* infected leaf tissue, showing paraphyses and urediospores highlighted in pseudo-color with orange, and leaf tissue in green, respectively (bottom right). **b** Transposable elements (TEs) content in different species of fungi (mostly plant pathogens), plants, and animals. The left histogram shows TEs proportion (%) per genome size, blue representing TEs content and grey non-TEs content; while the right histogram shows different classes of TEs in each genome. Source data are provided as a Source Data file.

formation, we can see a clear relaxation of repression during the *in planta* life stages of the pathogen. Due to the nested TEs, it is not possible at present to correlate specific TEs to specific expanded gene families. However, we can see that the *P. pachyrhizi* genome is expanded in genes related to amino acid metabolism and energy production, which may represent key lifestyle adaptations. Overall, our data unveil that TEs that started their proliferation during the radiation of the Leguminosae play a prominent role in the *P. pachyrhizi*'s genome and may have a key impact on a variety of processes such as host range adaptation, stress responses and plasticity of the genome. The high-quality genome assembly and transcriptome data presented here are a key resource for the community. It represents a critical step for further in-depth studies of this pathogen to develop new methods of control and to better understand the molecular dialogue between *P. pachyrhizi* and its agriculturally relevant host, Soybean.

## Results and discussion

### Two superfamilies of transposons dominate the *P. pachyrhizi* genome

The high repeat content and dikaryotic nature of the *P. pachyrhizi* genome poses challenges to genome assembly methods[15]. Recent improvements in sequencing technology and assembly methods have provided contiguous genome assemblies for several rust fungi[16–21]. Here, we have expanded the effort and provided reference-level

**Table 1 | *P. pachyrhizi* genome assembly metrics**

|                                      | K8108    | MT2006   | UFV02    |
|--------------------------------------|----------|----------|----------|
| Assembly size (Gb)                   | 1.083    | 1.0574   | 1.273    |
| Total no of contigs                  | 6505     | 7464     | 3140     |
| Contig N50 length (Kb)               | 278.753  | 222.464  | 677.464  |
| Max contig length (Mb)               | 3.028    | 3.054    | 4.158    |
| Min contig length (Kb)               | 16.399   | 21.118   | 11.733   |
| Complete BUSCOs (%)                  | 90.19    | 90.14    | 89.91    |
| Complete single-copy BUSCO (%)       | 15.70    | 15.87    | 22.56    |
| Complete duplicated BUSCO (%)        | 74.49    | 74.26    | 67.35    |
| Fragmented BUSCO (%)                 | 1.36     | 1.36     | 1.19     |
| Missing BUSCO (%)                    | 8.45     | 8.50     | 8.90     |
| Total BUSCO                          | 1764     | 1764     | 1764     |

genome assemblies of three *P. pachyrhizi* isolates (K8108, MT2006, and UFV02) using long-read sequencing technologies. All three isolates were collected from different regions of South America. We have used PacBio sequencing for the K8108 and MT2006 isolates and Oxford Nanopore for the UFV02 isolate to generate three high-quality genomes (Supplementary Fig. 1). Due to longer read lengths from Oxford nanopore, the UFV02 assembly is more contiguous compared to K8108 and MT2006 and is used as a reference in the current study (Table 1). The total genome assembly size of up to 1.25 Gb comprising two haplotypes, makes the *P. pachyrhizi* genome one of the largest fungal genomes sequenced to date (Fig. 1b). Analysis of the TE content in the *P. pachyrhizi* genome indicates ~93% of the genome consist of repetitive elements, one of the highest TE contents reported for any organism to date (Fig. 1b and Supplementary Data 1). This high TE content may represent a key strategy to increase genetic variation in *P. pachyrhizi*[22]. The largest class of TEs are class 1 retrotransposons, that account for 54.0% of the genome. The class II DNA transposons content is 34.0% (Supplementary Data 1 and 2). This high percentage of class II DNA transposons appear to be present in three lineages of rust fungi, the Melampsoraceae (*Melampsora larici-populina*), Pucciniaceae (*Puccinia graminis* f. sp. *triciti*) and Phakopsoraceae (*P. pachyrhizi*) (Fig. 1b). The recently assembled large genome (haploid genome size, 1 Gb) of the rust fungus *Austropuccinia psidii* in the family Sphaerophragmiaceae, however seems to mainly have expanded in retrotransposons[23]. This illustrates that TEs exhibit different evolutionary tracjectories in different rust taxonomical families. Over 80% of the *P. pachyrhizi* genome is comprised of only two superfamilies of TEs: long terminal repeat (LTR) and terminal inverted repeat (TIR) (Fig. 1b, and Supplementary Data 2). The largest single family of TE are the Gypsy retrotransposons comprising 43% of the entire genome (Fig. 2a, and Supplementary Data 2).

To understand the evolutionary dynamics of the different TE families present in the *P. pachyrhizi* genome, we compared the sequence similarities of TEs with their consensus sequences in the three genomes, which ranges from 65 to 100% sequence identity (Supplementary Fig. 2). Based on the concept of burst and decay evolution of TEs, the extent of sequence similarity between each TE copy to its cognate consensus is proportional to the divergence time of copies[24]. This approach allows us to compare within-genome relative insertion ages of TE insertions using consensus of TE families, a proxy for the ancestral sequence. TEs were categorised as (1) conserved TEs (copies with more than 95% identity), (2) intermediate TEs (copies with 85 to 95% identity) and (3) divergent TEs (copies with less than 85% identity)[24]. The average TE composition of the three isolates is 13.2–18.3% conserved, 29.4–29.9% intermediate and represent 51.7–57.3% divergent (Supplementary Fig. 3, and Supplementary Data 3-5). The average Gypsy retrotransposon composition of the three isolates is 16.5–20.7% conserved, 30.4–31.03% intermediate, and

48.8–52.5% divergent (Supplementary Fig. 3, and Supplementary Data 3-5). Similarly, average TIR composition of the three isolates is 12.2–18.4% conserved, 29.0–29.7 % intermediate and 51.8–57.8% divergent (Supplementary Fig. 3, and Supplementary Data 3-5). This suggests that i) multiple waves of TE proliferation have occurred during the history of the species, ii) the invasion of the two major TE families into the *P. pachyrhizi* genome is not a recent event, and iii) the presence of conserved TEs indicates ongoing bursts of expansion of TEs in the *P. pachyrhizi* genome. Therefore, the proportion and distribution of TEs indicate that different categories of TEs differentially shaped the genomic landscape of *P. pachyrhizi* during different times in its evolutionary history (Fig. 2b).

We set out to date the Gypsy and Copia TEs in *P. pachyrhizi*, using a TE insertion age estimation[25,26]. We observe that most retrotransposon insertions were dated less than 100 million years ago (Mya). We, therefore, decided to perform a more granulated study taking 1.0 million year intervals over this period. We approximated the start of TEs expansion at around 65 Mya after which the TE content gradually accumulates (Fig. 2c). We can see a more rapid expansion of TEs in the last 10 Mya, indeed over 40% of the Gypsy and Copia TEs in the genome seem to have arisen between today and 5 Mya (Fig. 2c). The climatic oscillations during the past 3 Myr are well known as the period of differentiation for multiple species[27]. Therefore, the genome expansion through waves of TE proliferation in *P. pachyrhizi* correlates with periods in which other species, including their host species the legumes started their main radiation, and differentiation due to external stressors[24–27]. This suggests that TEs either play an important role in generating the variation needed to adaptation of various stressors and/or proliferation of TEs is triggered by stressful events. Although a clear causal and or mechanistic role of TEs in adaptation, like in many other systems is still lacking[28,29], it is clear TEs have had a major impact on the architecture of the *P. pachyrhizi* genome.

## A subset of TEs is highly expressed during early *in planta* stages of infection

To build a high-quality resource that can facilitate future in-depth analyses, within the consortium, we combined several robust, independently generated RNAseq datasets from all three isolates that include major soybean infection-stages and in vitro germination (Fig. 3a, b). Altogether, eleven different stages are captured with seven having an overlap of two or more isolates, representing a total of 72 different transcriptome data sets (Fig. 3c). These data were used to support the prediction of gene models with the de novo annotation pipeline of JGI MycoCosm[30]. Those proteins secreted by the pathogen that impact the outcome of an interaction between host and pathogen are called effectors and are of particular interest[31,32]. We used a variety of complementary methods to identify 2,183, 2,027, and 2,125 secreted proteins (the secretome) encoded within the genome assembly of K8108, MT2006 and UFV02, respectively[33–37] (Supplementary Data 6-8). This is a two-fold improvement when compared to previous transcriptomic studies[38–42]. In *P. pachyrhizi*, depending on methodology, 36.73 – 42.30% of these secreted proteins are predicted to be effectors (Supplementary Data 6-8). We identified 437 common secreted proteins (shared by at least two isolates) that are differentially expressed at least in one time-point *in planta*, of which 246 are predicted to be effectors providing a robust set of proteins to investigate in follow-up functional studies (Supplementary Fig. 4, and Supplementary Data 9).

We performed expression analysis on the annotated TEs and observed that 6.66–11.65% of TEs are expressed in the three isolates (Supplementary Data 10 and 11). We compared the TE expression from different infection stages versus in vitro stages (Fig. 2a, and Supplementary Data 12-14) and used the *in planta* RNAseq data from the isolates K8108 and UFV02. A relatively small subset of TEs (0.03 – 0.25%)

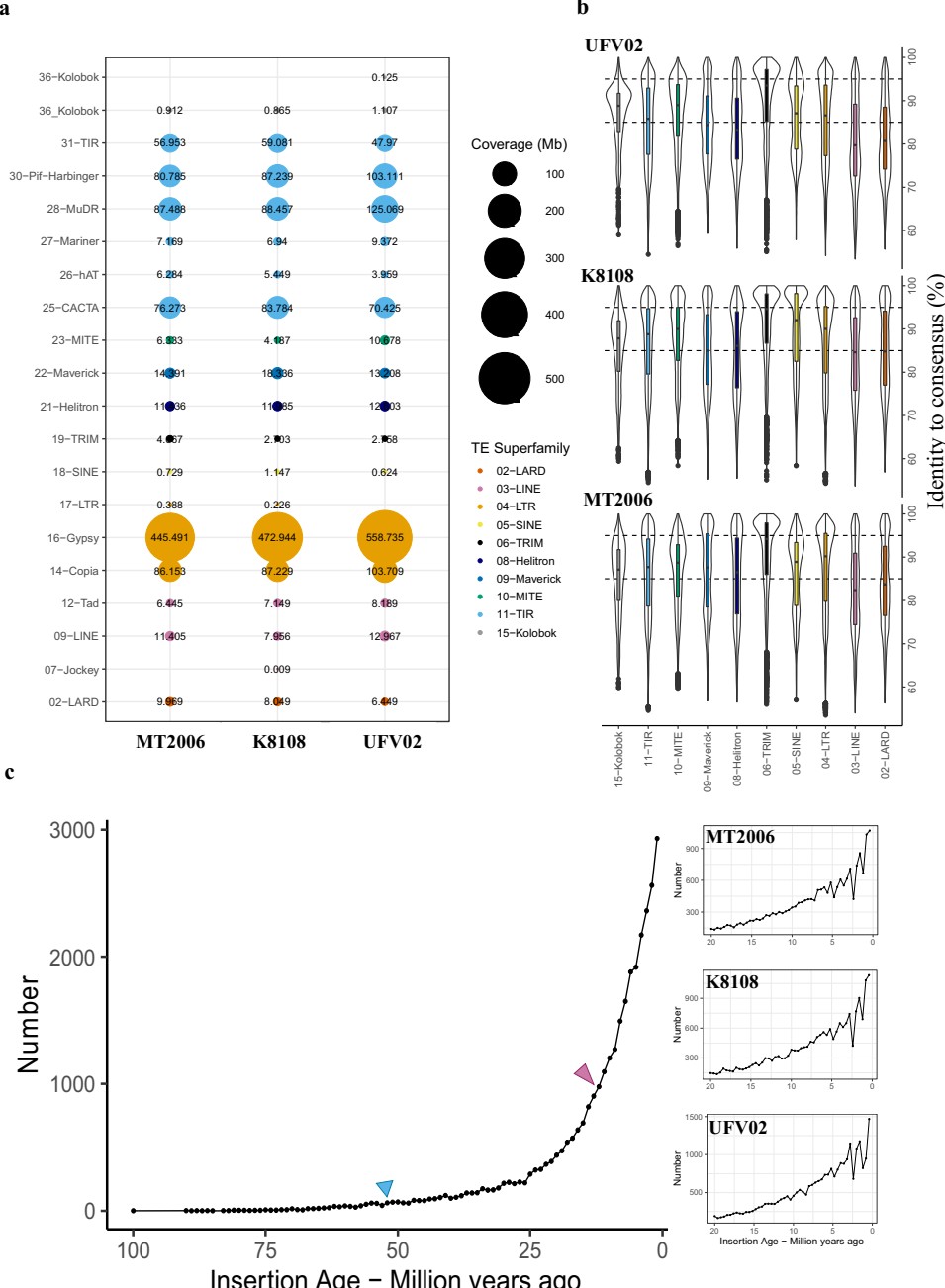

**Fig. 2 | Transposable element superfamilies in the *P. pachyrhizi* genomes, K8108, MT2006 and UFV02. a** Genome coverage of different TE superfamilies in three *P. pachyrhizi* genomes. **b** TE superfamilies are categorized based on the consensus identity, (1) conserved TEs, copies with more than 95% identity (2) intermediate TEs, copies with 85 to 95% identity and (3) divergent TEs, copies with less than 85% identity. Violin plots indicate: vertical line represents distribution at Q1-1.5 × IQR and Q3 + 1.5 × IQR, dots represent independent data points, first quartile (lower bar), median (thick line), third quartile (upper bar), and the shape

indicates the frequency. (n= one independent biological sample). **c** The number of LTR retrotransposons in UFV02 based on the insertion age (Million years ago, Mya) with 1.0 million year intervals (left). The legume speciation event around 53 Mya showed in blue triangle and -13 Mya whole genome duplication event in *Glycine* spp. marked with pink triangle[23]. In the right, the three plot shows recent burst of TEs between 0-20 Mya in three genomes of MT2006, K8108 and UFV02, respectively (n= one independent biological sample). Source data are provided as a Source Data file.

are expressed during the early infection stages between 10 to 72 hours post-inoculation (HPI) (Supplementary Fig. 5 and 6, and Supplementary Data 12-14). Remarkably, for this subset, we observed a 20 to 70-fold increase in the expression when compared to the spore and germinated-spore stages, with the expression levels reaching a peak at 24 HPI (Supplementary Fig. 5 and 6). To estimate the impact of the insertion age of this *in planta*-induced TE subset, we performed expression analysis on the conserved, intermediate, and divergent TEs. Although there is a slight overrepresentation of the conserved TEs,

several intermediate TEs and divergent TEs are also highly expressed during 10–24 HPI (Supplementary Fig. 7).

To compare the expression profile of this subset of TEs to the predicted effectors, we used the 246 core effectors and compared these with 25 known and constitutively expressed housekeeping genes across three isolates. We found that both TE and effector expression peaked at 24 HPI (Fig. 3d). While expression of effectors remained higher than the 25 selected housekeeping genes during infection, expression of TEs started to be repressed after 72 HPI (Fig. 3d). This

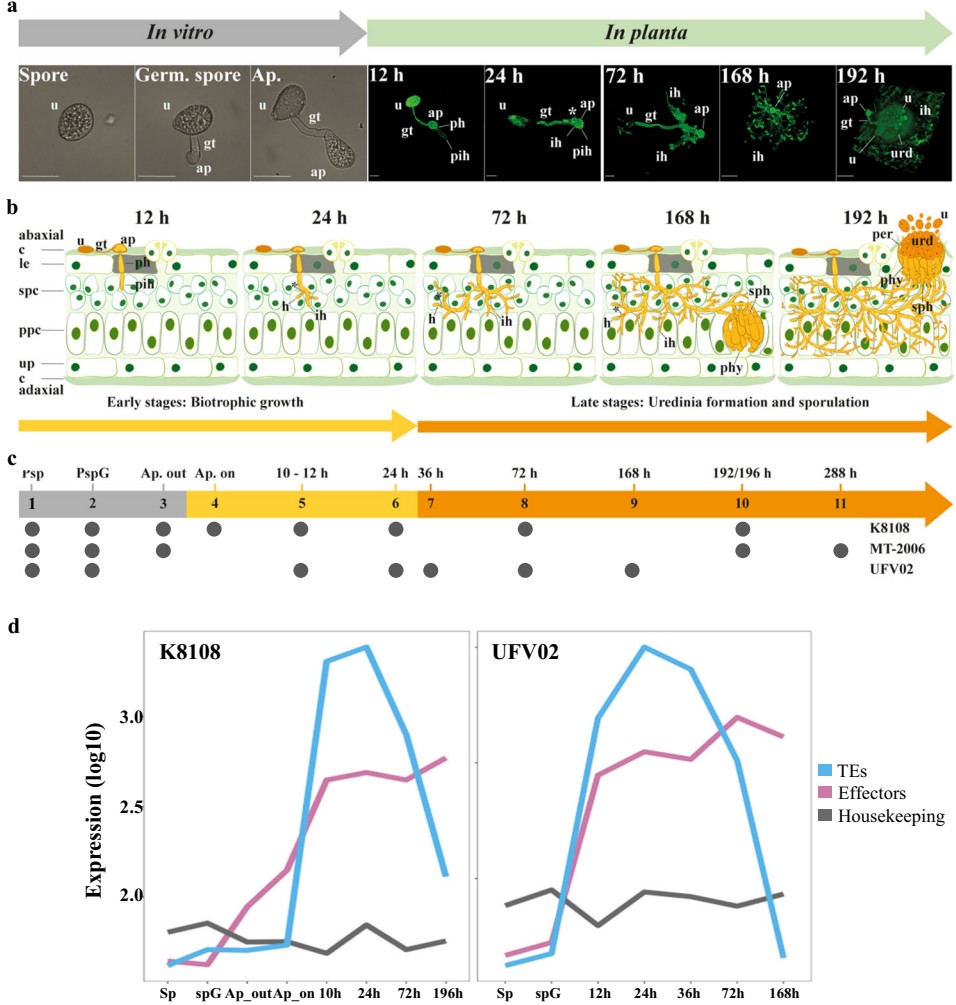

**Fig. 3 | Infection cycle of *P. pachyrhizi* and gene expression on the critical infection stages. a** Developmental phases of *P. pachyrhizi* infection in vitro and *in planta* on susceptible soybean plants. Scale bar, 10 μm for the in vitro germinated assays micrograph, 20 μm for the 12 h – 72 h *in planta* micrograph, and 50 μm for 168 h and 192 h *in planta* micrograph. Representative micrographs are shown from three independently performed assays with similar results. **b** Schematic of critical infection stages shown in the panel (a). **c** RNA sequencing on the critical time-points from three isolates. The timepoints included in this study are indicated by grey circles for each isolate. **d** Average expression (log10 of CPM) of the 246 core effectors and expressed TEs (Supplementary Data 10 and 11) compared to the housekeeping genes during different stages of infection in K8108 and UFV02 isolates. (n= three independent biological replicates). Source data are provided as a Source Data file. **Abbreviations:** urediospores (u), germ tube (gt), appressorium (ap), penetration hypha (ph), primary invasive hypha (pih), haustorial mother cell (*), haustorium (h), invasive hyphae (ih), sporogenous hyphae (sph), paraphyses (phy), peridium (per), uredinium (urd), cuticle (c), lower epidermal cells (lec), spongy parenchyma cells (spc), palisade parenchyma cells (ppc), upper epidermal cells (ue).

observation would corroborate the hypothesis of stress-driven TE de-repression observed in other patho-systems[43–45]. However, it also shows that in *P. pachyrhizi* only a small percentage of the TEs are highly expressed during early infection stages.

In several different phytopathogenic species a distinct genomic organization or compartmentalization can be observed for effector proteins. For example, the bipartite genome architecture of *Phytophthora infestans* and *Leptospheria maculans* in which gene sparse, repeat-rich compartments allow rapid adaptive evolution of effector genes[46]. Other fungi display other organizations such as virulence chromosomes[47,48] or lineage-specific regions[49,50]. However, when interrogating both genomic location and genomic distribution of the predicted candidate effector genes in *P. pachyrhizi*, we could not detect an analogous type of organization (Supplementary Fig. 8a-c). In addition, we did not observe evidence of the specific association between TE superfamilies and secreted protein genes (Supplementary Fig. 9), as has been observed in other fungal species[46,48,51–53]. Additional analyses comparing the distance between BUSCO (Benchmarking

Universal Single-Copy Orthologue) genes and genes encoding secreted proteins also showed no specific association (Supplementary Fig. 8d). Therefore, despite the large genome size and high TE content of *P. pachyrhizi*, its genome appears to be organized in a similar fashion to other rust fungi with smaller genome sizes[17,18,23,54]. The lack of detection of a specific association between TE and genes in *P. pachyrhizi* may be due to the level of TE invasion with 93% TE observed for this genome.

## *P. pachyrhizi* in South America is a single lineage with high levels of heterozygosity

Rust fungi are dikaryotic, therefore variation can exist both between isolates and between the two nuclei present in each cell of a single isolate. Long-term asexual reproduction is predicted to promote divergence between alleles of loci[55], which in principle can increase indefinitely[56]. Some rusts can reproduce both sexually and asexually leading to a mixed clonal/sexual reproduction. In the rust fungus *P. striiformis* f.sp. *tritici*, asexual lineages showed a higher

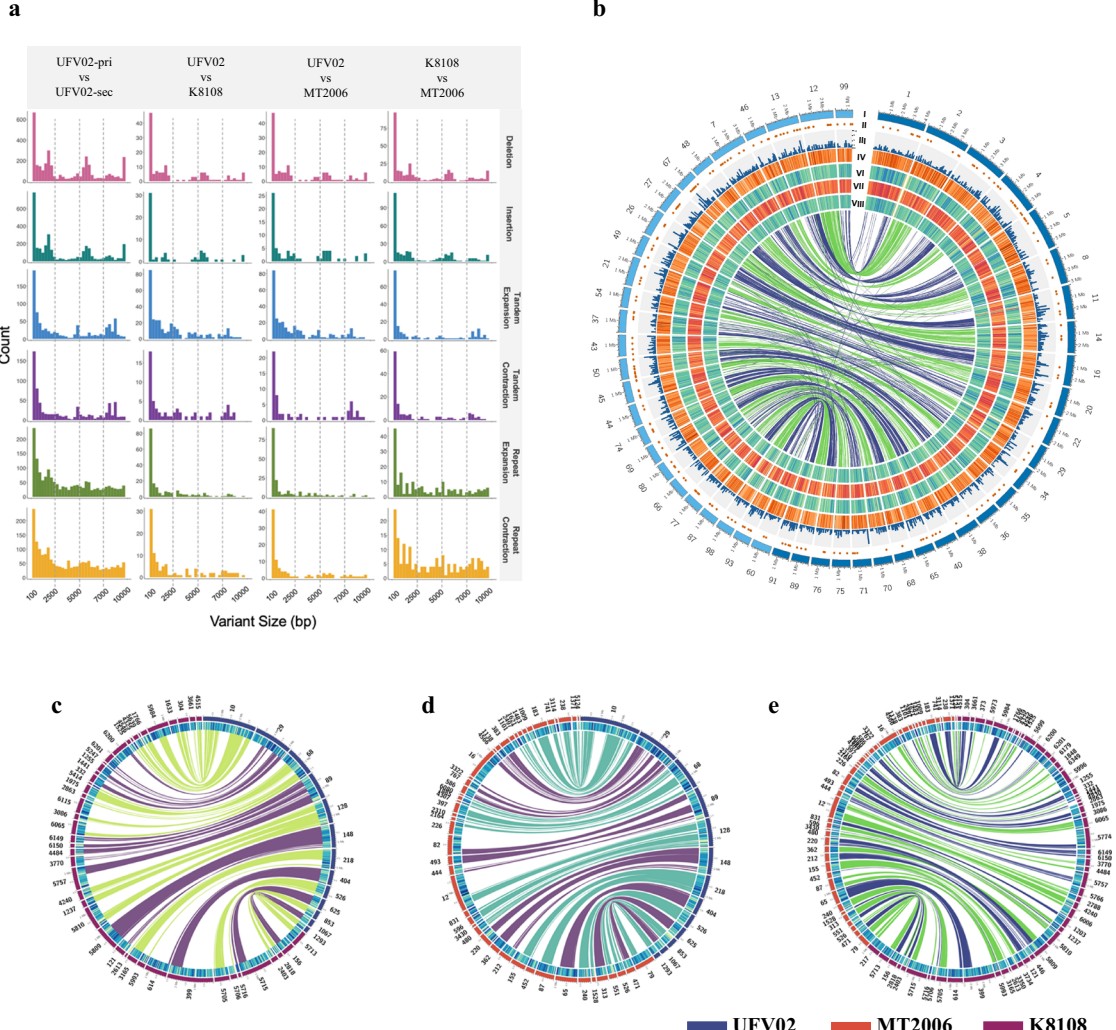

**Fig. 4 | Structural variation between *P. pachyrhizi* haplotypes is higher than variation between isolates. a** Density plots with different structural variation between haplotypes and across isolates. **b** Circos plot representing inter-haplotype variation in the isolate UFV02 . Layers from outside: **I** dark blue represent primary haplotigs and light blue secondary haplotigs; **II** secreted protein; **III** gene density (100 kb); **IV** TE density (50 kb); **V** SNP density K8108 isolate (25 kb); **VI** SNP density MT2006 isolate; **VII** SNP density UFV02 isolate (25 kb). **c-e** Circos plot showing inter-isolate variation. Layers from outside: **I** contigs from isolates represent in different colors; **II** TE density. Source data are provided as a Source Data file.

degree of heterozygosity between two haploid nuclei when compared to the sexual lineages[57]. In the case of *P. pachyrhizi*, there are clear indications that the population is propagating asexually in South America based on early studies using simple-sequence repeats (SSR) and internal transcribed spacer (ITS) sequences[58,59]. Our data utilizing high coverage raw Illumina data corroborate these earlier studies as we observed high levels of heterozygosity; 2.47% for UFV02, 1.61% for K8108 and 1.43% in MT2006, respectively (Supplementary Fig. 1a). This was further corroborated by mapping the Illumina reads to the genome assembly. In total, 283.355, 359.939, and 458.719 SNPs were identified from K8108, MT2006 and UFV02, respectively. The average heterozygous SNPs across the genome is 2.97 SNPs per Kb in UFV02 compared to 2.58 and 3.34 SNPs per Kb in K8108 and MT2006, respectively (Supplementary Data 15).

We subsequently studied the structural variation (insertions and deletions, repeat expansion and contractions, tandem expansion and contractions) as well as the haplotype variation between the three isolates (Supplementary Data 16)[60]. Remarkably, the structural variation between the haplotypes of UFV02 is 163.3 Mb, while the variation between the complete genomes of the three isolates is 8 to 13 Mb (Fig. 4a). For example, the total number of repeat expansion and

contractions is 7 and 16 times higher between the haplotypes than the variation between the isolates (Fig. 4a). To look at this inter-haplotype variation in more detail, we selected contigs larger than 1 Mb to study large syntenic blocks between isolates and haplotigs. The largest of these contigs, the 1.3 Mb contig 148 from UFV02 has synteny with contig 5809 from K8108, and contigs 220 and 362 from MT2006 (Fig. 4c-e), but not with its haplotig genome counterpart within UFV02, which indicates lack of recombination between haplotypes. This corroborates earlier studies that in South America *P. pachyrhizi* reproduces only asexually[61].

Collection of the monopustule isolates K8108, MT2006, UFV02 is separated in both time and geographical location (i.e. K8108 from *Colonia*, Uruguay, 2015; MT2006 from *Mato Grosso do Sul*, Brazil, 2006; UFV02 from *Minas Gerais*, Brazil, 2006). To study SNP variation, we mapped the Illumina data of all three isolates to the reference assembly of UFV02. Given the high level of heterozygosity and TE content, we focused our analysis on the now annotated exome space (Supplementary Data 15a). After removal of SNPs shared between either all three or two of the isolates, we identified only three non-synonymous mutations unique for K8180, eight non-synonymous mutations for MT2006 and five unique non-synonymous mutations for

**Table 2 | Expansion of gene families in the *P. pachyrhizi* genome**

| | Piwi | KOG0573 | KOG1481 | KOG2410 | KOG0399 | KOG2467 | KOG0683 | KOG2617 | KOG1261 | KOG1494 |
|---|---|---|---|---|---|---|---|---|---|---|
| *P. pachyrhizi* UFV02 | 531 | 78 | 28 | 62 | 48 | 12 | 10 | 15 | 26 | 13 |
| *P. pachyrhizi* MT2006 | 568 | 77 | 25 | 22 | 44 | 8 | 5 | 12 | 29 | 8 |
| *P. pachyrhizi* K8108 | 608 | 74 | 34 | 78 | 18 | 11 | 8 | 11 | 24 | 13 |
| *C. quercuum* f. sp. *fusiforme* G11 | 3 | 1 | 2 | 3 | 2 | 1 | 3 | 2 | 1 | 2 |
| *M. larici-populina* | 3 | 1 | 2 | 2 | 2 | 5 | 4 | 2 | 1 | 3 |
| *M. allii-populina* 12AY07 | 6 | 1 | 3 | 3 | 2 | 1 | 5 | 2 | 1 | 2 |
| *P. graminis* f. sp. *tritici* | 3 | 1 | 2 | 2 | 2 | 2 | 3 | 2 | 1 | 2 |
| *P. striiformis* f. sp. *tritici* 104 E137 A- | 7 | 2 | 5 | 4 | 2 | 4 | 8 | 4 | 3 | 4 |
| *P. coronata avenae* 12SD80 | 5 | 2 | 4 | 2 | 8 | 4 | 5 | 5 | 2 | 2 |
| *P. triticina* 1-1 BBBD Race 1 | 3 | 2 | 3 | 2 | 1 | 2 | 5 | 2 | 1 | 2 |

UFV02. For these 16 predicted genes, we found evidence for expression in our transcriptome analyses for ten genes. This total number of non-synonymous mutations within exons between the isolates may appear counterintuitive given the time and space differences between collection of these isolates. Nonetheless, it is likely that other single pustule isolates identified from another field would yield a similar number of mutations. Approximately 6 million spores may be produced per plant in a single day resulting in $3 \times 10^{12}$ spores per hectare per day[62]. Therefore, the ability to generate variation through mutation cannot be underestimated. We observed an enrichment of mutations in the upstream and downstream regions of protein-coding genes (Supplementary Data 15b), similar to other rust fungi[63–65]. In contrast to the low number of mutated exons, the number of uniquely expressed genes between the three isolates is relatively high when compared to the core set of differentially expressed genes (Supplementary Data 17-19). This may reflect a mechanism in which transcriptional variation is generated via modification of promotor regions which would have the advantage that coding sequences that are not beneficial in a particular situation can be "shelved" for later use. This would result in a set of differentially transcribed genes for different isolates, and a core set of genes that are transcribed in each isolate.

## The *P. pachyrhizi* genome is expanded in genes related to amino acid metabolism and energy production

We subsequently set out to identify expanding and contracting gene families within *P. pachyrhizi*. To this end, a phylogenetic tree of 17 selected fungal species (Supplementary Data 20a) was built using 408 conserved orthologous markers. We estimated that *P. pachyrhizi* diverged from its most recent common ancestor 123.2–145.3 million years ago (Supplementary Fig. 10 and Supplementary Data 20b), a time frame that coincides with the evolution of the Pucciniales[66,67]. We derived gene families including orthologues and paralogues from a diverse set of plant-interacting fungi and identified gene gains and losses (i.e. family expansions and contractions) using computational analysis of gene family evolution (CAFÉ) (Supplementary Data 20a)[68]. Genomes of rust fungi including *P. pachyrhizi* underwent more extensive gene losses than gains, as would be anticipated for obligate biotrophic parasites (Supplementary Fig. 11). In total, we identified 2,366 contracted families and 833 expanding families within UFV02, including 792 and 669 families with PFAM domains, respectively. The most striking and significant contraction in the *P. pachyrhizi* genome is related to DEAH helicase which is involved in many cellular processes, e.g., RNA metabolism and ribosome biogenesis (Supplementary Data 21). In contrast, significant expansions in 12 gene families were found, including genes encoding glutamate synthase, GMC (glucose-methanol-choline) oxidoreductase and CHROMO (CHRromatin Organisation MOdifier) domain-containing proteins (Supplementary Data 22). Glutamate synthase plays a vital role in nitrogen metabolism, and its ortholog in the ascomycete *Magnaporthe oryzae MoGLT1* is

required for conidiation and complete virulence on rice[69]. GMC oxidoreductase exhibits important auxiliary activity 3 (AA3_2) according to the Carbohydrate-Active enzymes (CAZy) database[70] and is required for the induction of asexual development in *Aspergillus nidulans*[71]. An extensive approach was used for the global annotation of CAZyme genes in *P. pachyrhizi* genomes, and after comparison with other fungal genomes, we also found clear expansions in glycoside hydrolases (GH) family 18 and glycosyltransferases (GT) family 1 (Supplementary Data 23). GH18 chitinases are required for fungal cell wall degradation and remodelling, as well as multiple other physiological processes, including nutrient uptake and pathogenicity[72,73].

The Phakopsoraceae to which *P. pachyrhizi* belongs represents a new family branch in the order Pucciniales[1]. With three *P. pachyrhizi* genome annotation replicates available, next to the above CAFÉ-analysis, we can directly track gene family expansions and contractions in comparison to genomes previously sequenced. We, therefore, compared *P. pachyrhizi* to the taxonomically related families Coleosporiaceae, Melampsoraceae and Pucciniaceae, which in turn may reveal unique lifestyle adaptations (Table 2).

The largest uniquely expanded gene family (531-608 members) in *P. pachyrhizi* comprises sequences containing the Piwi (P-element Induced Wimpy testes in Drosophila) domain (Table 2). Typically, the Piwi domain is found in the Argonaute (AGO) complex, where its function is to cleave ssRNA when guided by dsRNA[74]. Interestingly, classes of longer-than-average miRNAs known as Piwi-interacting RNAs (piRNAs) that are 26-31 nucleotides long are known in animal systems. In *Drosophila*, these piRNAs function in nuclear RNA silencing, where they associate specifically with repeat-associated small interfering RNA (rasiRNAs) that originate from TEs[75]. As in other fungal genomes, the canonical genes coding for large AGO proteins with canonical Argonaute, PAZ and Piwi domains can be observed in the genome annotation of the three *P. pachyrhizi* isolates. The hundreds of expanded predicted Piwi genes consist of short sequences of less than 500 nt containing only a partial Piwi domain aligning with the C-terminal part of the Piwi domain in the AGO protein. Some of these genes are pseudogenes marked by stop codons or encoding truncated protein forms, while others exhibit a partial Piwi domain starting with a methionine and eventually exhibiting a strong prediction for an N-terminal signal peptide. These expanded short Piwi genes are surrounded by TEs, several hundreds of which, but not all, are found in close proximity to specific TE consensus identified by the REPET analysis in the three *P. pachyrhizi* isolates (e.g. Gypsy, CACTA and TIR; Supplementary Fig. 12). However, no systematic and significant association could be made due to the numerous nested TEs present within the genome[76]. Moreover, none of the expanded short Piwi domain genes are expressed in the conditions we tested. However, in many systems, Piwis and piRNAs play crucial roles during specific developmental stages where they influence epigenetic, germ cell, stem cell, transposon silencing, and translational regulation[77]. Finally, the

domain present in these short Piwi genes is partial, and we do not know whether they retain any RNase activity. Therefore, we cannot validate at this stage the function of this family, which warrants further study and attention as it may represent either a new type of TE-associated regulator within *P. pachyrhizi*, or an expansion of a control mechanism to deal with this highly repetitive genome.

Several families related to amino acid metabolism have expanded greatly when compared to the respective families in other rust fungi, most notably Asparagine synthase (KOG0573), which has ~75 copies in *P. pachyrhizi* compared to two copies in Pucciniaceae and one copy in Melampsoraceae (Table 2). Similarly, expanded gene families can be observed in citrate synthase (KOG2617), malate synthase (KOG1261), NAD-dependent malate dehydrogenase (KOG1494). These enzymes are involved in energy production and conversion via the citrate cycle required to produce certain amino acids and the reducing agent NADH (Table 2). Next to the molecular dialogue with effector proteins, plant-pathogen interactions are a "tug-of-war" of resources between the host and the pathogen[78]. A key resource to secure in this process is nitrogen, a raw material needed to produce proteins. Therefore, the expansion in amino acid metabolism may reflect an adaptation to become more effective at securing this resource. Alternatively, the expanded categories also may reflect the metabolic flexibility needed to facilitate the broad host range of *P. pachyrhizi*, which to date comprises 153 leguminous species in 56 genera[13].

Associations with TEs are often a sign for adaptive evolution as they facilitate the genetic leaps required for rapid phenotypic diversification[44,79–81]. Gene duplication and gene family expansion can be directly linked to transposition activity due to imprecise excision and re-insertions and carry other genetic sequences[82]. Transposition-independent mechanisms may also promote structural rearrangements leading to gene family expansions through the recombination of homologous regions between TE copies. The TEs in these expansions may potentially be inactive[82]. We, therefore, investigated whether the expansion in amino acid metabolism could reflect a more recent adaptation by studying the TEs in these genomic regions. Furthermore, as described above, a distinction can be made between more recent bursts of TE activity (high conservation of the TEs) and older TE bursts leading to degeneration of the TE sequence consensus[83]. However, despite the presence of several copies of specific TE subfamilies (i.e. related to the same annotated TE consensus) in the vicinity of the surveyed expanded families such as amino acid metabolism, CAZymes and transporter related genes (Supplementary Fig. 13 and 14), no significant enrichment could be observed for any particular TE when compared to the overall TE content of the genome. This may reflect the challenge of making such clear associations due to the continuous transposition activity, which results in a high plasticity of the genomic landscape and a highly nested TE structure. Alternatively, it may suggest a more ancient origin of these expansions that have subsequently been masked by repetitive episodes of relaxed TE expression (Supplementary Fig. 15 and 16).

## Methods
### Fungal strain and propagation
*P. pachyrhizi* isolates, K8108, MT2006 and UFV02[84] are single uredosoral isolates collected from Uruguay (*Colonia* in 2015), Brazil (*Mato Grosso do Sul* in 2006) and Brazil (*Minas Gerais* in 2006), respectively. The isolates were propagated on susceptible soybean cultivars Abelina, Thorne, Toliman and Williams 82 by spraying a suspension of urediospores 1 mg ml$^{-1}$ in 0.01 % (vol/vol) Tween-20 in distilled water onto 21-day-old soybean plants followed by 18 h incubation in an incubation chamber at saturated humidity, and at 22 °C in the dark. Infected plants were kept at 22 °C, 16-h day/8-h night cycle and 300 μmol s$^{-1}$ m$^{-2}$ light. After 14 DPI (days post-inoculation), the pustules were formed, and the urediospores were harvested using a Cyclone surface sampler (Burkard Manufacturing Co. Ltd.) and stored at

−80 °C. The genomic DNA extraction methods are explained in Supplementary methods.

### Genomic DNA extraction and genome sequencing
The high molecular weight (HMW) genomic-DNA was extracted using a carboxyl-modified magnetic bead protocol[85] for K8108, a CTAB-based extraction for MT2006[86], and a modified CTAB protocol for UFV02[87].

For K8108, a 20-kb PacBio SMRTbell library was prepared by Genewiz (South Plainfield, NJ) with 15-kb Blue Pippin size selection being performed prior to sequencing on a PacBio Sequel system (Pacific Biosciences, Menlo Park, CA). The K8108 PacBio Sequel genomic reads yielding 69 Gbp of sequence data were error corrected using MECAT[88]; following parameter optimization for contiguity and completeness, the longest corrected reads yielding 50x coverage were assembled with MECAT's mecat2canu adaptation of the Canu assembly workflow[89], using an estimated genome size of 500 Mbp and an estimated residual error rate of 0.02. The resulting assembly had further base pair-level error correction performed using the Arrow polishing tool from PacBio SMRTTools v5.1.0.26412[90].

MT2006 genome was sequenced using the Pacific Biosciences platform. The DNA sheared to >10 kb using Covaris g-Tubes was treated with exonuclease to remove single-stranded ends and DNA damage repair mix, followed by end repair and ligation of blunt adapters using SMRTbell Template Prep Kit 1.0 (Pacific Biosciences). The library was purified with AMPure PB beads and size selected with BluePippin (Sage Science) at >6 kb cutoff size. PacBio Sequencing primer was then annealed to the SMRTbell template library, and sequencing polymerase was bound to them using Sequel Binding kit 2.0. The prepared SMRTbell template libraries were then sequenced on a Pacific Biosystem's Sequel sequencer using v2 sequencing primer, 1 M v2 SMRT cells, and Version 2.0 sequencing chemistry with 1 × 360 and 1 × 600 sequencing movie run times. The *Phakopsora pachyrhizi* MG2006 v1.0 genome was sequenced with PacBio, assembled with MECAT, polished with arrow, and annotated with the JGI Annotation Pipeline.

For UFV02, the PromethION platform of Oxford nanopore technology (ONT) (Oxford, UK) was used for long-read sequencing at KeyGene N.V. (Wageningen, The Netherlands). The libraries with long DNA fragments were constructed and sequenced on the PromethION platform. The raw sequencing data of 110 Gbp was generated and was base-called using ONT Albacore v2.1 available at https://community.nanoporetech.com. The UFV02 genome assembly, the longest 15, 20, 25, 30, 34, 40 and 56x nanopore reads were assembled using the Minimap2 and Miniasm pipeline[91]. To improve the consensus, error correction was performed three times with Racon using all the nanopore reads[92]. The resulting assembly was polished with 50x Illumina PCR-free 150 bp paired-end reads mapped with bwa[93] and Pilon[94], and repeated three times. We assessed the BUSCO scores after each step to compare the improvement in the assemblies.

### Genome annotation
The gene predictions and annotations were performed in the *P. pachyrhizi* genomes K8108, MT2006 and UFV02 in parallel using the JGI Annotation Pipeline[30]. TE masking was done during the JGI procedure, which detects, and masks repeats and TEs. Later, the extensive TE classification performed with REPET was imported and visualized as a supplementary track onto the genome portals. RNAseq data from each isolate was used as intrinsic support information for the gene callers from the JGI pipeline. The gene prediction procedure identifies a series of gene models at each gene locus and proposes the best gene model to define a filtered gene catalogue. Translated proteins deduced from gene models are further used for functional annotation according to international reference databases. All the annotation information is collected into an open public JGI genome portal in the MycoCosm (https://mycocosm.jgi.doe.gov/Phakopsora) with dedicated tools for

community-based annotation[30,95]. In total, 18,216, 19,618 and 22,467 gene models were predicted from K8108, MT2006 and UFV02, respectively (Supplementary Data 24); of which 10,492, 10,266 and 9,987 genes were functionally annotated. We have performed differential expression analyses using the germinated spores as a reference point in each of the three isolates (Supplementary Fig. 17, and Supplementary Data 17-19). A total of 3,608 common differentially expressed genes (DEGs) were identified in at least one condition shared between two or more isolates (Supplementary Fig. 18, and Supplementary Data 25).

## Quality assessment of the whole-genome assemblies

The whole-genome assemblies of *P. pachyrhizi* were evaluated using two different approaches. First, we used BUSCO version 5.0[96] to assess the genic content based on near-universal single-copy orthologs with basidiomycetes_odb10 database, including 1764 gene models. Second, K-mer's from different assemblies were compared using KAT version 2.4.1[97]. Genome heterozygosity was estimated using Genome-Scope 2.0[98].

## Insertion age of LTR-retrotransposons

Full-length LTR-retrotransposons were identified from the *P. pachyrhizi* genomes using LTRharvest with default parameters, and this tool belongs to the GenomeTools genome analysis software v1.6.1[99]. LTRs annotated as Gypsy or Copia were used for molecular dating, and selection was based on a BLASTX against Repbase v20.11[100]. 3′ and 5′ LTR sequences were extracted and aligned with mafft v7.471[101], and alignments were used to calculate Kimura's 2 P distances[102]. The insertion age was determined using the formula $T = K / 2r$, with K the distance between the 2 LTRs and r the fungal substitution rate of $1.05 \times 10^{-9}$ nucleotides per site per year[25,26].

## Molecular dating and Phylogenetic analysis

The phylogenetic tree was generated after the alignment of 408 conserved orthologous markers identified from at least 13 out of 17 genomes using PHYling (https://github.com/stajichlab/PHYling_unified). The sequences were aligned and concatenated into a super-alignment with 408 partitions. The phylogenetic tree was built with RAxML-NG (v0.9.0) using a partitioned analysis, and 200 bootstraps replicates. Molecular dating was established with mcmctree from PAML v4.8. Calibration points were extracted to Puccinialies[67] and Sordariomycetes–Leotiomycetes[103]. The 95% highest posterior density (HPD) values are calibrated to the node.

## Sample preparation for RNAseq

For expression analysis, 11 different stages were evaluated, with eight stages having an overlap of two or more isolates. These stages were nominated 1-11, as illustrated in Fig. 3c. For K8108, seven in vitro, one *on planta* and eight in planta samples, each with three biological replicates, were generated and used to prepare RNA libraries. To get in vitro germ tubes and fungal penetration structures, a polyethylene foil (dm freezer bag, Karlsruhe, Germany) was placed in glass plates and inoculated with a spore suspension (2 mg ml⁻¹). Each biological replicate corresponded to 500 cm² foil and ~4 mg urediospores. The plates were incubated at 22 °C in the dark at saturated humidity for 0.5, 2, 4 or 8 h. After incubation, the spores were collected using a cell scraper. For the appressoria-enriched sample, urediospore concentration was doubled and the plates rinsed with sterile water after 8 h of incubation prior to collection. The material was ground with mortar and pestle in liquid nitrogen. The time 0.5 h was considered as spore (Spore, Psp - stage 1), the 2 h as a germinated spore (Germinated spore, PspG – stage 2), and the 8 h rinsed as appressoria enriched sample in vitro (stage 3). The samples of spores collected after 4 and 8 h were not used for expression analysis. To obtain *on planta* fungal structures, three-week-old soybean plants (Williams 82) were

inoculated as mentioned above. After 8 HPI, liquid latex (semi-transparent low ammonium, Latex-24, Germaringen, Germany) was sprayed (hand spray gun with gas unit, Preval, Bridgeview, USA) until complete leaf coverage. After drying off, latex was removed. It contained the appressoria and spores from the leaf surface but no plant tissue. This sample was considered as enriched in appressoria on plant and is exclusive for K8108 isolate (stage 4). Three middle leaflets of different plants were bulked for each sample and ground in liquid nitrogen using a mortar and pestle. The inoculated leaf samples were harvested at 10, 24, 72 and 192 HPI (stages 5, 6, 8 and 10) for the *in planta* gene expression studies.

For MT2006, the germ tubes, and appressorium were produced on polyethylene (PE) sheets where urediospores were finely dusted with household sieves held in a double layer of sifting. The PE sheets were then sprayed with water using a chromatography vaporizer and were kept at 20 °C, 95% humidity in the dark. For germ tubes the structures were scratched from the PE sheets after 3 h (stage 2) and for appressoria after 5 h (stage 3). The formation of both germ tubes and appressoria was checked microscopically. The in vitro samples were only used when there were at least 70% germ tubes or appressoria. The structures were dried by vacuum filtration and stored in 2-ml microcentrifuge tubes at −70 °C after freezing in liquid nitrogen. The resting spores came directly from storage at −70 °C (stage 1). For the *in planta* samples, 21 days old soybean cultivar Thorne was sprayed with a suspension containing 0.01% Tween-20, 0.08% milk-powder and 0.05% urediospores. The inoculated plants were kept, as mentioned previously. The samples were taken using a cork borer (18 mm diameter) at 192 and 288 HPI (stages 10 and 11). Three leaf pieces were collected for each sample (three times and from three different plants) for every time-point, stored in liquid nitrogen and kept at −80 °C.

For UFV02, the spore suspension of 1×10⁶ spores ml⁻¹ concentration was prepared in 0.01% v/v Tween-20. Four weeks old soybean plants were sprayed thoroughly on the abaxial surface of the leaves, and the plants were kept at saturated humidity in the dark for 24 h. After 24 h, plants were kept at 22 °C and 16/8-h light/dark cycle. The leaf samples were collected from non-inoculated plants (0 h) and infection-stages at 12, 24, 36, 72 and 168 HPI (stages 5, 6, 7, 8 and 9). An infection assay was performed in three biological replicates, and three plants were used for each replicate. All the samples were stored in liquid nitrogen after collection and kept at −80 °C for further processing (stage 1). Spores were harvested after 14 days post-inoculation and used for the RNA extraction. The urediospores were germinated in vitro on the water surface in a square petri dish and kept for 6 h at 24 °C (stage 2). The germinated-urediospores were collected in a falcon tube and snap freeze in liquid nitrogen. The samples were freeze-dried and kept at −80 °C until further processing. The un-inoculated plants (0 h) were not used in the expression analysis.

## RNA isolation and sequencing

All the samples were ground in liquid nitrogen, and the total RNA was extracted using the Direct-zol RNA Miniprep Plus Kit (ZymoResearch, Freiburg, Germany), the mirVana™ miRNA Isolation Kit (Ambion/life technologies, Calsbad, CA, USA), and TRIzol™ reagent (Invitrogen) according to the manufacturer's protocols for K8108, MT2006, and UFV02, respectively. The quality of RNA was assessed using the TapeStation instrument (Agilent, Santa Clara, CA) or the Agilent 2100 bioanalyzer.

The RNA libraries from K8108 were normalized to 10 mM, pooled, and sequenced at 150-bp paired-end on the HiSeq X instrument at Genewiz (South Plainfield, NJ), with ten samples per lane. The transcriptome of MT2006 was sequenced with Illumina. Stranded cDNA libraries were generated using the Illumina Truseq Stranded mRNA Library Prep Kit. mRNA was purified from 1 ug of total RNA using magnetic beads containing poly-T oligos. mRNA was fragmented and reversed transcribed using random hexamers and SSII

(Invitrogen) followed by second-strand synthesis. The fragmented cDNA was treated with end-pair, A-tailing, adapter ligation, and 8 cycles of PCR. The prepared libraries were quantified using KAPA Biosystem's next-generation sequencing library qPCR kit (Roche) and run on a Roche LightCycler 480 real-time PCR instrument. The quantified libraries were then multiplexed, and the pool of libraries was prepared for sequencing on the Illumina HiSeq sequencing platform utilising a TruSeq paired-end cluster kit, v4, and Illumina's cBot instrument to generate a clustered flow cell for sequencing. Sequencing of the flow cell was performed on the Illumina HiSeq 2500 sequencer using HiSeq TruSeq SBS sequencing kits, v4, following a 2×150 indexed run recipe. The RNA samples of UFV02 were sequenced at the Earlham Institute (Norwich, UK) on Illumina HiSeq 2500 platform with 250-bp paired-end reads. Eight different samples (as mentioned above) in three biological replicates were used for the RNA library preparation. All 24 libraries were multiplexed and sequenced on six lanes of HiSeq 2500.

## TE analysis
The TE insertions are categorised based on the sequence identity 1) TEs with less than 85% sequence identity to the consensus, called old insertions, 2) TEs with 85-95% sequence identity are intermediate, and 3) TEs with more than 95% identity represent recent insertions (Supplementary Fig. 2 and 3)[24]. All three isolates show common patterns of consensus identity, and a majority of the TEs show an intermediate age of insertions (Supplementary Fig. 2). The retrotransposon superfamilies such as terminal-repeat retrotransposons in Miniature (TRIMs) are the most recent expansion and long interspersed nuclear element (LINE), and large retrotransposon derivative (LARD) superfamilies are the most ancient insertion in the *P. pachyrhizi* genome (Supplementary Fig. 3). To verify the relationship between secreted genes and TEs, we calculated the distance between these features using Bedtools[104] with Closest algorithm, which returns the smallest genomic distance between two features. From the results obtained, we calculated the number of TEs neighbouring each secreted gene, grouped them by each TE superfamily and built the graphs. The tools used for analysis and graphs construction were Pandas v.1.3.4 and Seaborn 0.11.2 libraries, together with Python 3.9.7.

## Identification of assembly haplotigs
The haplotypes were phased using the purge-haplotig pipeline[105] using Illumina WGS data. The haplotigs were aligned with their corresponding primary contigs using Mummer-4.0 for UFV02[106]. Assemblytics was subsequently used to define six major types of structural variants[60], including insertions and deletions, repeat expansion and contractions, and tandem expansion and contractions.

The assembly was compared to itself using blastn (NCBI-BLAST + 2.7.1) with max_target_seqs = 10 and culling_limit = 10. After filtering for sequences matching themselves, overlaps among the remaining high-scoring segment pairs (HSPs) of > = 500 bp and > = 95% identities were consolidated with an interval tree requiring 60% overlap, then chained using MCScanX_h[107] to determine collinear series of matches, requiring three or more collinear blocks and choosing as a candidate haplotig sequences having at least 40% of their length subsumed by a chain corresponding to a longer contig sequence. For downstream analyses requiring a single haplotype representation, hard masking was applied to remove overlapped regions from the haplotigs using BEDtools v2.27.0[104]. To identify gene correspondence among the three isolates, we used Liftoff software[108]. The genome assembly of each isolate was used as a reference to map the other two isolates' gene catalogue with >95% coverage and identity of >95%. The correspondence was established based on the gene annotation coordinates of each reference genome and the mapping coordinates from liftoff results (Supplementary Data 26).

## Read mapping, variant calling and SNP effect prediction
Illumina paired-end reads of the three isolates were trimmed with Trimmomatic v0.36[109] to remove adapters, barcodes, and low-quality sequences with the following parameters: illuminaclip = TruSeq3-PE-2.fa:2:30:10, slidingwindow = 4:20, minlen = 36. Then, sequence data from all three isolates were aligned to the reference assembly of *P. pachyrhizi* UFV02 v2.1 using BWA version 0.7.17 with the BWA-mem algorithm[93], with the options -M -R. Alignment files were converted to BAM files using SAMtools v1.9[110], and duplicated reads were removed using the Picard package (https://broadinstitute.github.io/picard/). The GATK v3.8.1 software[111] was used to identify and realign poorly aligned reads around InDels using Realigner Target Creator and Indel Realigner tools, creating a merged bam file for all the three isolates. The subsequent realigned BAM file was used to calling SNPs and InDels using HaplotypeCaller in GATK and filtering steps were performed to kept only high-quality variants, as following: the thresholds setting as: "QUAL < 30.00 || MQ < 40.00 || SOR > 3.00 || QD < 2.00 || FS > 60.00 || MQRankSum < −12.500 || ReadPosRankSum < −8.00 || ReadPosRankSum > 8.00". The resulting SNPs and InDels were annotated with snpEffect v4.1[112].

## Infection and disease progression
*P. pachyrhizi* is an obligate biotrophic fungus which forms a functional appressorium to penetrate the host epidermal layer within 12 HPI (hours post-inoculation)[113]. The penetrated epidermal cell dies after fungus establishes the penetration hyphae (PH) and forms the primary invasive hyphae (PIH) in the mesophyll cells after 24 HPI (Fig. 3a, b). The PIH differentiates and forms a haustorial mother cell, establishing the haustorium in the spongy parenchyma cells. At 72 HPI, the fungus colonises the spongy and palisade parenchyma cells (spc and ppc)[114] (Fig. 3a, b). At 168 HPI, the uredinium starts to develop in the palisade parenchyma. At 196 HPI, the epidermal layer is broken, and the fully developed uredinia emerge. Each pustule forms thousands of urediospores and carries on the infection (Supplementary Fig. 19).

## RNA transcriptome assembly
The low-quality RNA-seq reads were processed and trimmed using Trimmonatic version 0.39[109] with the parameters ILLUMINA-CLIP:2:30:10 LEADING:3 HEADCROP:10 SLIDINGWINDOW:4:25 TRAILING:3 MINLEN:40 and read quality was assessed with FastQC version 0.11.5 (https://www.bioinformatics.babraham.ac.uk/projects/fastqc/). The high quality reads were filtered for any potential contamination among the fungi reads using Kraken2 software and parameter −unclassified-out for soybean genome and any possible contaminant species[115]. After all filtering steps, reads from each library were mapped against the three isolates assemblies using STAR v2.7.6a[116]. Parameters for mapping were (--outSAMtype BAM SortedByCoordinate, --outFilterMultimapNmax 100, --outFilterMismatchNmax 2, --outSAMattrIHstart 0, --winAnchorMultimapNmax 200, and −outWigType bedGraph). After mapping, duplicated reads were removed using Picard v.2.23.2. Htseq was used to count reads and Deseq2 to identify the differentially expressed genes in appressorium or during the host colonization relative to expression levels in the germinated-spore condition.

To validate gene annotation dedup-BAM files were analysed using StringTie v2.1.2[117], and the gtf files obtained were merged (-m 600 -c 5) for genes and (-m 200 -c 5) for TE (TE). The final gtf file was compared with each genome annotation file per isolate using gffcompare[118] software to validate the annotate genes and TEs. We detected 18,132, 19,467, and 22,347 genes presenting transcriptional evidence in K8108, MT2006 and UFV02 genomes, respectively, demonstrating high sensitivity (>93.9%) and precision in a locus level (>75.4%) in all three isolates (Supplementary Fig. 20 and Supplementary Data 27). For functional annotation, genes were considered expressed when each transcriptome reads were mapped against its respective reference

genome, considering the criteria of TPM (Transcripts Per Kilobase Million) values > 0 in at least two biological replicates.

The BAM-dedup files obtained as above described were applied for TE expression analyses using TEtranscript software[119]. TE read counts were normalised between replicates in different conditions using R/Bioconductor package EdgeR v.3.1[120,121]. Only TEs with a minimum of one read in at least two replicates were considered in this normalisation step. Libraries were normalised with the TMM method[122], and CPM (counts per million) were generated with the EdgeR v.3.13. To better understand the expression distribution of TEs in the K8108, MT2006 and UFV02 genomes, we constructed boxplot plots to visualize the variation of expression values (average CPM) in each of their conditions. For this, we calculated the arithmetic means, the standard deviation, and the quartile values of the TEs expression in each condition for the isolates K8108, MT2006 and UFV02.

### Prediction and annotation of secreted proteins

To predict classically secreted proteins, we initially searched for proteins containing a classic signal peptide and no transmembrane signal using SignalP (versions 3 and 5)[36], TMHMM[123] and Phobius[124] programs. For the identification of additional secreted proteins without a classic peptide signal and no transmembrane signal (non-classically secreted), we used EffectorP (versions 1 and 2)[33,34] and TMHMM programs. In both approaches, we kept the proteins having a TM in the N-term region. The proteins selected by both approaches were analysed by PS-SCAN program[125] to remove putative endoplasmic reticulum proteins. All programs were performed considering default parameters. The secreted proteins predicted in the previous step were annotated using Blast[126], RPSBlast, PredGPI[127], InterProScan[128] and hmmsearch[129] programs. Similarity searches using Blast program were performed against the NCBI non-redundant (nr), FunSecKb[130], Phi-base[131], and LED[132] databases, applying an e-value of $10^{-5}$. To search for domains in sequences, we used the programs RPSBlast and hmmsearch against the Conserved Domain Database (CDD)[133] and PFAM database[134], respectively, using an e-value of $10^{-5}$ in both cases. Orthologue mapping was done through similarity searches with the hmmsearch program against profile HMMs obtained from eggNOG database[135]. To predict the localisation of proteins in the cellular compartments, ApoplastP[136], Localizer[137], targetP[138], WoLFPSORT[139], and DeepLoc[140] programs were used using default parameters. To assign a final localisation for each protein, the following criteria were considered: if at least two programs found the same result, that result was considered as a predicted location. Otherwise, the term "Not classified" was assigned to the protein. To identify the motifs [Y/F/W]xC in the sequences, we used a proprietary script developed in Perl language. A summary of the prediction and annotation pipelines for the secreted proteins is illustrated in Supplementary Fig. 21 and 22.

For the prediction of putative effector proteins, we used the list of predicted secreted proteins containing a classical signal peptide. For the prediction of candidate effector proteins in each genome, we defined three different approaches. In the first one, sequences predicted as "Extracellular" or "Not Classified" by the location programs and with no annotation were selected as candidates for effector proteins. We obtained 618, 531 and 598 candidates to effector proteins in K8108, MT2006 and UFV02 with this approach. In the second approach, we selected proteins with PFAM domains present in effector proteins[141–152]. Applying this criterion, we selected 142, 128 and 55 candidates in K8108, MT2006 and UFV02, respectively. Finally, in the third approach, we ran EffectorP program to classify the effector candidates, and we obtained 802, 851 and 899 candidates in K8108, MT2006 and UFV02 genomes, respectively (Supplementary Data 6-8).

### Staining of leaf samples and microscopy

Plants were inoculated by spray inoculation, and leaves were harvested at the indicated time points. Samples were destained in 1 M KOH with 0.01% Silwet L-77 (Sigma Aldrich) for at least 12 h at 37 °C and stored in 50 mM Tris-HCl pH 7.5 at 4 °C. Fungal staining was obtained with wheat germ agglutinin (WGA) FITC conjugate (Merck L4895), samples were incubated 30 min to overnight in a 20 μg/ml solution in Tris-HCl pH 7.5. Co-staining of plant tissue with propidium iodide (Sigma-Aldrich P4864) was performed according to the manufacturer's instructions. Images were obtained with a Leica SP5 confocal microscope (Leica Microsystems) with an excitation of 488 nm and detection at 500-550 nm and 625-643 nm, respectively. Z-stacks were opened in the 3D viewer of the LAS X software (Leica Application Suite X 3.5.7.23225), and the resulting images were exported. Clipping was performed as indicated in the pictures. Shading was performed in some cases for better visualisation.

For cryo-scanning electron microscopy, inoculated soybean leaves were cut and mounted on an aluminium stub with Tissue Tek OCT (Agar Scientific Ltd, Essex, UK) and plunged frozen in slushed liquid nitrogen to cryo-preserve the material before transfer to the cryo-stage of a PP3010 cryo-SEM preparation system (Quorum Technologies, Laughton, UK) attached to a Zeiss Gemini 300 field emission gun scanning electron microscope (Zeiss UK Ltd, Cambridge, UK). Surface frost was sublimated by warming the sample to −90 °C for 4 minutes before the sample was cooled to −140 °C and sputter coated with platinum for 50 seconds at 5 mA. The sample was loaded onto the cryo-stage of the main SEM chamber and held at −140 °C during imaging at 3 kV using an Everhart-Thornley detector. False colouring of images was performed with Adobe Photoshop 22.4.2.

### Reporting summary

Further information on research design is available in the Nature Portfolio Reporting Summary linked to this article.

### Data availability

Source data are provided with this paper. The raw sequencing data of MT2006, K8108 and UFV02 isolates has been deposited at NCBI under the accession numbers PRJNA368291, PRJEB46918, and PRJEB44222, respectively. Source data are provided with this paper.

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

## Acknowledgements

We thank Dan MacLean, Christian Schudoma, and Ram Krishna Shrestha for bioinformatics support. Bioinformatics infrastructure was supported in part by NBI Research Computing. We thank Matthew Moscou and Michael C. Schatz for many fruitful discussions. We acknowledge Heike Popovitsch for technical support. We thank Robert Dietrich and Lucio Garcia (Syngenta RTP) for their technical support with the sequencing of K8108 genome and transcriptome.

The work (Proposal 10.46936/10.25585/60000959) conducted by the U.S. Department of Energy Joint Genome Institute. (https://ror.org/04xm1d337), a DOE Office of Science User Facility, is supported by the Office of Science of the U.S. Department of Energy under Contract No. DE-AC02-05CH11231.

Sequencing and RNAseq analyses of UFV02 was supported by 2Blades.

Work on the soybean isolate K8108 in the Conrath and Schaffrath lab was supported, in part, by Syngenta Crop Protection.

EM, CL and SD were in part funded by the Labex Arbre (Programme Investissement d'Avenir, ANR-11-LABX-0002-01).

## Author contributions

Y.K.G, F.C.M.G., C.L., A.F., S.H., E.G.C.F, V.S.L., L.S.O., E.M., S.W., Co.C., Y.I., K.T., K.R., E.D., B.H., K.L., A.M.R.B., E.P., V.S., Ch.D., Cé.D., M.v.H, A.J., L.C., Y.T., J.R., B.d.V.A.M., A.W., H.S., S.P., L.G.Z., V.C.H., F.C., T.I.L., D.B., A.M., S.K., S.B., L.W., Ci.C, M.Y., M.C.M, Q.L., M.L., S.H.B., and S.D. performed research. Y.K.G., C.L., A.F., S.H., E.G.C.F, V.S.L., L.S.O., A.M.R.B., E.M., S.W., C.C., Y.I., E.D., B.H., A.J., A.W., B.d.V.A.M., L.G.Z., T.I.L., M.L., S.H.B., and S.D. analyzed the data. Y.K.G., F.C.M.G., M.L, S.D., and H.P.v.E. edited the manuscript. Y.K.G., F.C.M.G., V.S.L., L.S.O., M.L., U.S., S.D., and H.P.v.E. wrote the paper. F.C.M.G., V.N., P.G., R.T.V., I.V.G., U.C., G.S., C.S., S.D., and H.P.v.E. directed aspects of the project.

## Competing interests

Connor Cameron, Andrew Farmer, Dirk Balmer, Stephanie Widdison, Qingli Liu and Gabriel Scalliet were employees of Syngenta or affiliates during the research project. Work on the soybean isolate K8108 in the Conrath and Schaffrath lab was supported, in part, by Syngenta Crop Protection. Yogesh Kumar Gupta, Everton Geraldo Capote Ferreira, Kelly Robinson, and H. Peter van Esse have a collaboration with Bayer crop science on Asian soybean rust.
The remaining authors declare no competing interests.

## Additional information

Yogesh K. Gupta [1,2], Francismar C. Marcelino-Guimarães [3], Cécile Lorrain [4], Andrew Farmer[5], Sajeet Haridas [6], Everton Geraldo Capote Ferreira [1,2,3], Valéria S. Lopes-Caitar [3], Liliane Santana Oliveira[3,7], Emmanuelle Morin[8], Stephanie Widdison[9], Connor Cameron[5], Yoshihiro Inoue[1,2], Kathrin Thor[1,2], Kelly Robinson[1,2], Elodie Drula [10,11], Bernard Henrissat [12,13], Kurt LaButti [6], Aline Mara Rudsit Bini[3,7], Eric Paget[14], Vasanth Singan[6], Christopher Daum[6], Cécile Dorme[14], Milan van Hoek[15], Antoine Janssen[15], Lucie Chandat[14], Yannick Tarriotte[14], Jake Richardson[16], Bernardo do Vale Araújo Melo[17], Alexander H. J. Wittenberg[15], Harrie Schneiders[15], Stephane Peyrard [14], Larissa Goulart Zanardo[17], Valéria Cristina Holtman[17], Flavie Coulombier-Chauvel[14], Tobias I. Link[18], Dirk Balmer[19],

André N. Müller[20], Sabine Kind [20], Stefan Bohnert[20], Louisa Wirtz[20], Cindy Chen[6], Mi Yan[6], Vivian Ng [6], Pierrick Gautier [14], Maurício Conrado Meyer [3], Ralf Thomas Voegele [18], Qingli Liu[21], Igor V. Grigoriev [6,22], Uwe Conrath [20], Sérgio H. Brommonschenkel[17], Marco Loehrer [20], Ulrich Schaffrath [20], Catherine Sirven[14], Gabriel Scalliet [19,23], Sébastien Duplessis [8,23] & H. Peter van Esse [1,2,23] ✉

[1]2Blades, Evanston, Illinois, USA. [2]The Sainsbury Laboratory, University of East Anglia, Norwich, UK. [3]Brazilian Agricultural Research Corporation - National Soybean Research Center (Embrapa Soja), Paraná, Brazil. [4]Pathogen Evolutionary Ecology, ETH Zürich, Zürich, Switzerland. [5]National Center for Genome Resources, Santa Fe, New Mexico, USA. [6]U.S. Department of Energy Joint Genome Institute, Lawrence Berkeley National Laboratory, Berkeley, California, USA. [7]Department of Computer Science, Federal University of Technology of Paraná (UTFPR), Paraná, Brazil. [8]Université de Lorraine, INRAE, IAM, Nancy, France. [9]Syngenta Jealott's Hill Int. Research Centre, Bracknell Berkshire, UK. [10]AFMB, Aix-Marseille Univ., INRAE, Marseille, France. [11]Biodiversité et Biotechnologie Fongiques, INRAE, Marseille, France. [12]Department of Biological Sciences, King Abdulaziz University, Jeddah, Saudi Arabia. [13]DTU Bioengineering, Technical University of Denmark, Kgs, Lyngby, Denmark. [14]Bayer SAS, Crop Science Division, Lyon, France. [15]KeyGene N.V., Wageningen, The Netherlands. [16]The John Innes Centre, Norwich, UK. [17]Departamento de Fitopatologia, Universidade Federal de Viçosa, Viçosa, Brazil. [18]Institute of Phytomedicine, University of Hohenheim, Stuttgart, Germany. [19]Syngenta Crop Protection AG, Stein, Switzerland. [20]Department of Plant Physiology, RWTH Aachen University, Aachen, Germany. [21]Syngenta Crop Protection, LLC, Research Triangle Park, Durham, NC, USA. [22]Department of Plant and Microbial Biology, University of California Berkeley, Berkeley, CA, USA. [23]These authors contributed equally: Gabriel Scalliet, Sébastien Duplessis, H. Peter van Esse. ✉e-mail: Peter.vanesse@tsl.ac.uk

