## [Peer Review File · Nature Communications]

Major proliferation of transposable elements shaped the genome of the soybean rust pathogen *Phakopsora pachyrhizi*REVIEWER COMMENTS

Reviewer #1 (Remarks to the Author):

In this manuscript authors present genome assemblies for three isolates of *Phakopsora pachyrhizi*, which causes Asian soybean rust. Authors have overcome a very significant technical hurdle to generate these assemblies due to the repetitive composition of the genome. As authors point out, this is a very important pathogen of soybean that can be a limiting factor in production unless it is managed, mostly with fungicides. The genome of this pathogen represents an important step in understanding its evolution, mechanisms for generating genetic diversity, and pathogenicity and virulence. The manuscript is focused on the very high percentage of transposable elements in the genome, secreted proteins and potential effectors, and some expanded gene families. The manuscript will benefit from some improvements for clarity.

Comments:

Table 2 – “*P. 13oronate avenae*” should be corrected

Line 831 – “bule” to “blue”

Line 150 – “(Fig 2C)..” delete extra “.”

Line 150 – “Strikingly, fossil records....” Would this sentence make more sense after the sentence on when TE expansion started in *P. pachyrhizi*. These two thoughts seem to logically flow together followed by the description of more recent events.

Line 841 – Recommend to change the wording to: “The timepoints included in this study are indicated by black circles for each isolate.”

Fig. 3D – Label X-axis. Is it time or growth stages?

Line 168 – Did authors also consider Elmore et al. 2020 De novo transcriptome of *Phakopsora pachyrhizi* uncovers putative effector repertoire during infection. *Physiol. Mol. Plant Pathol.* 110:101464?

Supp. Fig. 4: Is this a log fold change for all samples relative to germinated spores? Please indicate in legend or on the scale bar.

Supp. Fig. 4 legend: change wording to something like “differentially expressed genes encoding predicted secreted proteins”

Fig. 3D – Is this fold change relative to germinating spores or is it log₁₀ CPM? Y-axis needs a more descriptive label along with better description in the figure legend.

Line 189-190 – Its hard to interpret this without labeled X-axes. Should we expect the effector plot to match the pixel intensities in Supp. Fig. 4?

Line 268 – recommend to change to: “We estimated that *P. pachyrhizi* diverged from its most recent common ancestor...”

Supp. Fig. 14b – There is a typo in the y-axis label “Atransporters”

Reviewer #2 (Remarks to the Author):

The submitted manuscript by Gupta et al. details the assembly and characterization of the causal agent of soybean rust, *Phakapsora pachyrhizi*. This is an economically important pathogen, causing significant yield loss in global soybean production, and has also been challenging to assemble due to its large size, high repeat content, and dikaryotic nature. The genome assemblies, annotations, and time-course RNA-seq

appear to be of very high quality and will be a great resource. There are many interesting findings regarding in planta TE expression, expansion of Piwi domain containing proteins, and families of genes that may be important for nutrient acquisition. Follow-up research in these areas will further our understanding of soybean rust biology.

I have a few suggestions for the manuscript:

It would be good to add the data for *Austropuccinia psidii* to Fig 1b. This is because the current assembly size figure looks like *P. pachyrhizi* is quite an outlier, but this is due to what is being compared. The genome is discussed in the results, and also used to make the point that the ClassII TE expansion is not uniform across rust fungi. Adding this genome, or another like it, will provide a stronger visual connection to the results.

The results for Lines 133-137 are confusing. I read that averaged over the 3 genomes, ~6% of Gypsy-LTRs are >95% identical to consensus, ~9.6% are 85-95% identical to consensus, and ~16% are <85% identical to consensus. With only 3 classes of categorizations, why does the average class membership only account for ~32% of the elements? It seems to me that closer to 100% of the elements should be accounted for in how they break into the 3 classes. If this is some odd result of computing the average, another metric should be used. The same issue for the TIR data. I also do not follow the implied logic of lines 141-143. Why the specific emphasis on TEs in the "conserved and to lesser extent intermediate" identity classes?

The conclusions for lines 150-154 may or may not be true, but is honestly an untestable hypotheses. The authors present that LTRs rapidly expanded 5-10 Mya. This also overlaps a time when climate oscillations have been associated with documented speciation events, and the diversification of legumes. But this does not mean that TE expansions 'correlates' with radiation and adaptation of legumes. No correlation is presented and the events overlap over an incredibly long time period. It may just be coincidence or the TE expansion was in response to something unrelated to legume radiation. This comes up again in the conclusions section, (Lines 363-365) "expansion of TEs within the genome correlates with the radiation and speciation of the legumes". It is not clear how TEs actually impact "host range adaptation, stress responses and plasticity of the genome (lines 370-373)". There are not data in the manuscript to address this. Speculation about these points is fine, but I suggest it be stated as such, and less as clear fact.

In my opinion, the findings for the TE expansion, and their interpretation, need explicit description of the limitations. The described TE annotation pipeline is robust and it looks like a lot of time and care went into the analysis. Even so, there is a lot about TEs we do not understand. The models and assumptions about TE age assume random mutation accumulation. There is evidence that TEs can be co-opted and serve functions in many eukaryotic genomes. This affects assumptions of selection and therefore age. Also, fungi are known to have odd mechanisms for targeting TEs, some of which may induce DNA mutations (<https://www.ncbi.nlm.nih.gov/pmc/articles/PMC5607778/>). So, how individual TEs, or classes or rounds of bursts are affected by these processes is rather unknown. I suggest that the authors add some clear language that TE dating and inference has plenty of uncertainty. I would also suggest to mention that repeats can also have a significant impact not just because of actual transposition, but by creating homologous sequence that serves as template for repair of DNA breaks. This can result in significant genome variation in the absence of actual TE activity.

Minor comments-

It is very difficult to see the color correspondence between the TE superfamily legend and the bubble colors for Fig. 2A. The legend bubbles need to be bigger, or another indication of which Y-axis labels fall under which TE superfamily.

There is some left-over x-axis legend showing for the inserts at the right of figure 2c.

There are double and missing periods lines 150 and 154.

Reviewer #3 (Remarks to the Author):

Major points:

- Two of the three genomes were sequenced with first-generation Pacbio reads and one with Nanopore reads. This resulted in 3k-7k contigs, which is underwhelming given that HiC + HiFi technologies have been the gold standard in the field for at least the past 1-2 years. Additionally, I have concerns about the assembly qualities and downstream analysis and their conclusions. For example, the three genomes should have been polished with genomic Illumina reads. Long-read polishing alone will not correct these first-generation assemblies appropriately. The authors also used Albacore for basecalling, which was discontinued in 2019. I believe the base accuracy without short-read polishing and without high-accuracy basecalling could be low for UFV02. Lastly, the two PacBio assemblies were assembled with two completely different pipelines which can pose problems in downstream comparative analyses.
- The authors removed duplicated reads from their RNAseq analysis. This could also remove real counts and thus underestimate gene expression. Why was this done, were lots of PCR duplicates present?
- DESeq2: the authors talk about using 'the calibrator', I have never heard of this and wonder what it refers to?
- I struggle with the comparisons of TE contents between genomes that were annotated with very different pipelines, e.g. comparing REPET to RepeatMasker (see statement line 113).
- The authors claim that in the Nanopore assembly structural variation between haplotypes is very high. However, it could also be due to collapsed regions, incorrect base calls and Nanopore assembly errors. See Duan et al 2022 (Genome Biology) for caveats around Nanopore assemblies. I think the structural variation they are reported are mostly Nanopore assembly artefacts. Collapsed regions and Illumina SNPs per Mb could be reported to assess genome assembly accuracy.

Minor points:

- What are the raw read genome coverages for the three assemblies?
- The authors claim to have used the odb9 database for the BUSCO analysis, but shouldn't that be 1,335 genes? It should also report higher complete BUSCOs in the other rust assemblies.
- Regarding the LiftOff tool, why was such a stringent coverage/identity of 95% set?
- Please change 'effector genes' to 'putative effector genes'
- Supp Fig2: what is alpha?
- Supp Fig22: please use a more scientific layout for this plot, i.e. bar charts next to each other.
- Line 94: please change 'frustrated' to 'hampered' or similar
- Line 102: these references are outdated. There are now the Pt76 and Pca203 and Pst134 genomes
- Line 110: should be haplotypes instead of haplotigs

REVIEWER COMMENTS

Reviewer #1 (Remarks to the Author):

In this manuscript authors present genome assemblies for three isolates of *Phakopsora pachyrhizi*, which causes Asian soybean rust. Authors have overcome a very significant technical hurdle to generate these assemblies due to the repetitive composition of the genome. As authors point out, this is a very important pathogen of soybean that can be a limiting factor in production unless it is managed, mostly with fungicides. The genome of this pathogen represents an important step in understanding its evolution, mechanisms for generating genetic diversity, and pathogenicity and virulence. The manuscript is focused on the very high percentage of transposable elements in the genome, secreted proteins and potential effectors, and some expanded gene families. The manuscript will benefit from some improvements for clarity.

Comments:

Table 2 – “*P. coronate avenae*” should be corrected

This has been corrected.

Line 831 – “bule” to “blue”

This has been corrected.

Line 150 – “(Fig 2C)..” delete extra “.”

This has been corrected.

Line 150 – “Strikingly, fossil records....” Would this sentence make more sense after the sentence on when TE expansion started in *P. pachyrhizi*. These two thoughts seem to logically flow together followed by the description of more recent events.

We agree with the reviewer and changed the sentence (lines 158-164).

Line 841 – Recommend to change the wording to: “The timepoints included in this study are indicated by black circles for each isolate.”

This has been changed as per recommendation.

Fig. 3D – Label X-axis. Is it time or growth stages?

This has been changed. The X-axis refers to the different stages of infection.

Line 168 – Did authors also consider Elmore et al. 2020 De novo transcriptome of *Phakopsora pachyrhizi* uncovers putative effector repertoire during infection. *Physiol. Mol. Plant Pathol.* 110:101464?

We have compared the secreted proteins from three genomes with Elmore et al. 2020 dataset (1428 secreted proteins). After filtering the redundant sequences, 969 unique secreted proteins were identified, suggest two-fold increase in the number of secreted proteins in the current study. We have added this result, adjusted the findings in line 177 accordingly and included the reference.

Supp. Fig. 4: Is this a log fold change for all samples relative to germinated spores? Please indicate in legend or on the scale bar.

We have added a scale bar and indicated in the figure's legend "The scale bar shows log₂ fold-change."

Supp. Fig. 4 legend: change wording to something like "differentially expressed genes encoding predicted secreted proteins"

Indeed, we have now changed per recommendation.

Fig. 3D – Is this fold change relative to germinating spores or is it log₁₀ CPM? Y-axis needs a more descriptive label along with better description in the figure legend.

&

Line 189-190 – Its hard to interpret this without labeled X-axes. Should we expect the effector plot to match the pixel intensities in Supp. Fig. 4?

This has been changed. The X-axis refers to the different stages of infection. The core effectors used for the plot in Figure 3d are a subset of all the secreted proteins shown in Supp. Fig. 4.

In Supp. Fig 4, we have added the scale with log₂ FC and changed the figure legend (lines 250-253). We include the relative expression compared to the germinated spores is on a log₂ fold scale.

In Figure 3d, we have changed the figure legend (lines 869-871) to provide clarity. In this case, we used a log₁₀ CPM (counts per million) as it allowed plotting against the TEs that were analysed in a log₁₀ CPM scale.

Line 268 – recommend to change to: "We estimated that *P. pachyrhizi* diverged from its most recent common ancestor..."

This has been changed as recommended.

Supp. Fig. 14b – There is a typo in the y-axis label "Atransporters"

This has been changed as recommended.

Reviewer #2 (Remarks to the Author):

The submitted manuscript by Gupta et al. details the assembly and characterization of the causal agent of soybean rust, *Phakopsora pachyrhizi*. This is an economically important pathogen, causing significant yield loss in global soybean production, and has also been challenging to assemble due to its large size, high repeat content, and dikaryotic nature. The genome assemblies, annotations, and time-course RNA-seq appear to be of very high quality and will be a great resource. There are many interesting findings regarding in planta TE expression, expansion of Piwi domain containing proteins, and families of genes that may be important for nutrient acquisition. Follow-up research in these areas will further our understanding of soybean rust biology.

I have a few suggestions for the manuscript:

It would be good to add the data for *Austropuccinia psidii* to Fig 1b. This is because the current assembly size figure looks like *P. pachyrhizi* is quite an outlier, but this is due to what is being compared. The genome is discussed in the results, and also used to make the point

that the ClassII TE expansion is not uniform across rust fungi. Adding this genome, or another like it, will provide a stronger visual connection to the results.

We agree with the reviewer. Fig 1b is changed and we have included *Austropuccinia psidii* in the figure.

The results for Lines 133-137 are confusing. I read that averaged over the 3 genomes, ~6% of Gypsy-LTRs are >95% identical to consensus, ~9.6% are 85-95% identical to consensus, and ~16% are <85% identical to consensus. With only 3 classes of categorizations, why does the average class membership only account for ~32% of the elements? It seems to me that closer to 100% of the elements should be accounted for in how they break into the 3 classes.

We agree with the reviewer and have changed the sentence to make for clearer writing (lines 130-142).

If this is some odd result of computing the average, another metric should be used. The same issue for the TIR data. I also do not follow the implied logic of lines 141-143. Why the specific emphasis on TEs in the “conserved and to lesser extent intermediate” identity classes?

We agree with the reviewer. This was originally intended to reflect that the conserved TEs are still impacting the genome today. However, we can see how it does not add but detract from the main flow of the text.

The conclusions for lines 150-154 may or may not be true, but is honestly an untestable hypotheses. The authors present that LTRs rapidly expanded 5-10 Mya. This also overlaps a time when climate oscillations have been associated with documented speciation events, and the diversification of legumes. But this does not mean that TE expansions ‘correlates’ with radiation and adaption of legumes. No correlation is presented and the events overlap over an incredibly long time period. It may just be coincidence or the TE expansion was in response to something unrelated to legume radiation. This comes up again in the conclusions section, (Lines 363-365) “expansion of TEs within the genome correlates with the radiation and speciation of the legumes”. It is not clear how TEs actually impact “host range adaptation, stress responses and plasticity of the genome (lines 370-373)”. There are not data in the manuscript to address this. Speculation about these points is fine, but I suggest it be stated as such, and less as clear fact.

Thanks for the input. We took these analyses as far as we could, and indeed it is clear that the TE expansion correlates with geological times in which many stressors were present. Climate change, host range radiation, glaciation and others. We have adapted the text as follows to reflect this nuance (lines 149-164):

*“We set out to date the Gypsy and Copia TEs in *P. pachyrhizi*, using a TE insertion age estimation^{22, 23}. We observe that most retrotransposon insertions were dated less than 100 million years ago (Mya). We, therefore, decided to perform a more granulated study taking 1.0 million year intervals over this period. We approximated the start of TEs expansion at around 65 Mya after which the TE content gradually accumulates (Fig. 2c). We can see a more rapid expansion of TEs in the last 10 Mya, indeed over 40% of the Gypsy and Copia TEs in the genome seem to have arisen between today and 5 Mya (Fig. 2c). The climatic oscillations during the past 3 Myr are well known as the period of extremely rapid*

*differentiation of multiple species²⁸. Therefore, the rapid genome expansion through waves of TE proliferation in *P. pachyrhizi* correlates with periods in which other species, including their host species the legumes started their main radiation, and differentiation due to external stressors²⁴⁻²⁷. This suggests that TEs either play an important role in generating the variation needed to adaptation of various stressors and/or proliferation of TEs is triggered by stressful events. Although a clear causal and or mechanistical role of TEs in adaptation, like in many other systems is still lacking, it is clear TEs have had a major impact on the architecture of the *P. pachyrhizi* genome in this time period. “*

The title now reads:

Major proliferation of transposable elements in the last 10 million years has shaped the genome of the soybean rust pathogen *Phakopsora pachyrhizi*

We have modified the concluding remarks accordingly as well (see line 381 and 382)

In my opinion, the findings for the TE expansion, and their interpretation, need explicit description of the limitations. The described TE annotation pipeline is robust and it looks like a lot of time and care went into the analysis. Even so, there is a lot about TEs we do not understand. The models and assumptions about TE age assume random mutation accumulation. There is evidence that TEs can be co-opted and serve functions in many eukaryotic genomes. This affects assumptions of selection and therefore age. Also, fungi are known to have odd mechanisms for targeting TEs, some of which may induce DNA mutations (<https://www.ncbi.nlm.nih.gov/pmc/articles/PMC5607778/>). So, how individual TEs, or classes or rounds of bursts are affected by these processes is rather unknown. I suggest that the authors add some clear language that TE dating and inference has plenty of uncertainty. I would also suggest to mention that repeats can also have a significant impact not just because of actual transposition, but by creating homologous sequence that serves as template for repair of DNA breaks. This can results in significant genome variation in the absence of actual TE activity.

This reviewer is indeed correct in these comments, and we do not undermine the limitations of TE age measurement. We made additions of lines 130-134 and changes within lines 136-142 on page 4. Also, we do agree as well about the variety of processes which may be at play beyond the sole transposition activity, such as the repair of DNA breaks through the presence of homologous sequences in the genome. We assume these processes are evident since they are addressed in reviews and articles we cite about TEs however words of caution and accuracy are always helpfull for naive readers. Thus, we modified the text accordingly, notably regarding homologous recombination, on lines 354-359 on page 8.

“Gene duplication and gene family expansion can be directly linked to transposition activity due to imprecise excision and re-insertions and carry other genetic sequences [Ref added: <https://www.sciencedirect.com/science/article/pii/S0168952522000361>]. Transposition-independent mechanisms may also promote structural rearrangements leading to gene family expansions through the recombination of homologous regions between TE copies. The genes in these expansions can potentially be inactive [Ref added: <https://www.nature.com/articles/nrq.2015.25>]. “

On the other hand, mechanisms such as the RIP as pointed out by this reviewer, so far have not been found in rust fungi [see ref: <https://bmcbgenomics.biomedcentral.com/articles/10.1186/s12864-015-1347-1>]. Since this is peripheral to the current study, and in order to keep the manuscript focused, we did not add mention of such processes.

Minor comments-

It is very difficult to see the color correspondence between the TE superfamily legend and the bubble colors for Fig. 2A. The legend bubbles need to be bigger, or another indication of which Y-axis labels fall under which TE superfamily.

This has been changed as recommended.

There is some left-over x-axis legend showing for the inserts at the right of figure 2c.

We have removed UFV02 as it was confusing for the bigger plot.

There are double and missing periods lines 150 and 154.

Indeed there were, and this has been addressed.

Reviewer #3 (Remarks to the Author):

Major points:

- Two of the three genomes were sequenced with first-generation Pacbio reads and one with Nanopore reads. This resulted in 3k-7k contigs, which is underwhelming given that HiC + HiFi technologies have been the gold standard in the field for at least the past 1-2 years. We don't disagree that the development in sequencing technology and results that can be obtained by HiFi and HiC have advanced rapidly in 1-2 years. These are powerful technologies as evidenced by e.g. the chromosome level assemblies of the *Puccinia graminis* (177 Mb), *P. tritricina* (260 Mb) and *P. coronata f. sp. avenae* (208 Mb), genomes that are of a considerable smaller size than the *Phakopsora pachyrhizi* genome.

However, the main value in this paper is that three isolates were sequenced to a high standard using the same robust annotation pipeline established at JGI, setting a new standard for the community. Hence, we disagree the results are "underwhelming". Expecting the quality and breadth of analyses we provide here with the latest technology on a 1.28 Gb genome inevitably leads to a "Catch-22". What we present here is the culmination of several years of analyses and work that will further our understanding of rust biology and fungal biology in general.

Additionally, I have concerns about the assembly qualities and downstream analysis and their conclusions. For example, the three genomes should have been polished with genomic Illumina reads. Long-read polishing alone will not correct these first-generation assemblies appropriately. The authors also used Albacore for basecalling, which was discontinued in 2019. I believe the base accuracy without short-read polishing and without high-accuracy basecalling could be low for UFV02.

These concerns are unwarranted as "Albacore" and the newer "Guppy" perform similar in terms of accuracy metrics. Guppy is an order of magnitude faster (~ 1,500,000 bp/s vs ~

120,000 bp/s) due to its use of GPU acceleration (Wick *et al.*, 2019). In addition, the UfV02 genome was error-corrected with short 50X coverage 150 bp paired end Illumina reads (please see Supplementary methods, lines 41-44). The MT2006 and K8108 genomes are error-corrected with the raw PacBio data (mentioned in the Materials and Methods, Supplementary methods, line 33).

Lastly, the two PacBio assemblies were assembled with two completely different pipelines which can pose problems in downstream comparative analyses.

We thank the author for pointing this out. After some research, it turns out that although MT2006 was initially assembled with Falcon, results were not of sufficient quality. Subsequently, MT2006 was analysed with Mecat, polished with arrow and annotated by the JGI Annotation pipeline, identical to K8108. Therefore, the two PacBio genomes were assembled with the same method as indicated in the JGI portal. We have now addressed this error of documentation in the M&M (Supplementary methods, lines 32-34).

- The authors removed duplicated reads from their RNAseq analysis. This could also remove real counts and thus underestimate gene expression. Why was this done, were lots of PCR duplicates present?

The reasons that we have decided to remove the duplicated reads were:

We compared three isolates with several different sources of RNA-seq data that went through different sample preparation, completely independent library construction, and the number of PCR cycles. We can minimize the effects of PCR amplification biases by removing duplicates. This is important because:

- a. After Kraken filtering, the number of reads became more uneven among samples.
- b. To further validate gene annotation, we applied StringTie v2.1.2 using the parameter (-c 5) for minimal read coverage to identify real gene and determine gene structure. Therefore, PCR duplicates would overestimate the percentages of validated intron/exon structure.
- c. The most abundant genes show the lowest Coefficient of Variation and have the highest probability of duplicates (optical, amplification, or true duplicates) as these genes are over-represented in the RNAseq data. Most normalization protocols for the differential expression end up artificially inflating the variance for these genes and have impact on the conclusions from the DE analysis. In contrast, genes with low expression are mostly impacted by duplicates for the same reason (i.e. their variance is artificially reduced by normalization. In the raw sequencing data, duplicates have a larger proportion in the total count to begin with and thus increasing type I error).

- DESeq2: the authors talk about using 'the calibrator', I have never heard of this and wonder what it refers to?

We refer to the "calibrator" as the sample used as a reference or standard condition in comparison to apressorium or host infection to identify DEGs. To avoid confusion, we have now replaced the term "calibrator" with "expression relative to the germinated spore condition" (Supplementary methods, lines 146-148).

- I struggle with the comparisons of TE contents between genomes that were annotated

with very different pipelines, e.g. comparing REPET to RepeatMasker (see statement line 113).

This interpretation is not correct. A detailed annotation of TE has been performed in parallel on each of the three genomes applying strictly the same analysis process with the REPET pipeline, a tool extensively used in Repeat annotation in fungal genomes. As mentioned in the supplementary methods lines 47-67, RepeatMasker is part of the TEannot module of REPET.

- The authors claim that in the Nanopore assembly structural variation between haplotypes is very high. However, it could also be due to collapsed regions, incorrect base calls and Nanopore assembly errors. See Duan et al 2022 (Genome Biology) for caveats around Nanopore assemblies I think the structural variation they are reported are mostly Nanopore assembly artefacts.

We appreciate the comment. To accommodate readers that will have the same question, we have added raw read-depth coverage generated with Illumina HiSeq 150 bp PE plots for three genomes in Supplementary Fig 1c. Please note this display few reads with a 2x haploid coverage for all three genomes indicating only few collapsed regions. We modified the Supplementary Fig 1 legend accordingly at lines 224-226.

In addition, as mentioned above, Illumina 50x 150 bp PE error correction has been performed (Supplementary methods, lines 41-44). Therefore, the risk in our analyses for incorrect base calls is minimal.

To ensure the highest possible contiguity of the nanopore assembly we have considered only the longest 40X coverage nanopore reads for this analyses. The structural variation we observe was further corroborated with the synteny analysis, where we used contigs larger than 1 Mb. This way we identified clear syntenic blocks between the isolates compared to the haplotypes in UFV02 genome assembly.

Collapsed regions and Illumina SNPs per Mb could be reported to assess genome assembly accuracy.

We believe performing Illumina SNPs per Kb to assess genome assembly accuracy is an excellent suggestion. This will allow readers to make proper inferences and assumptions about the analyses we report on going forward. These are now indicated as supplementary table 15a.

Minor points:

- What are the raw read genome coverages for the three assemblies?

We have included these in Supplemental figure S1 as S1d

- The authors claim to have used the odb9 database for the BUSCO analysis, but shouldn't that be 1,335 genes? It should also report higher complete BUSCOs in the other rust assemblies.

Thanks for spotting the odb9, we used the odb10 database with 1764 genes. Hence the numbers presented are correct (line 436).

- Regarding the LiftOff tool, why was such a stringent coverage/identity of 95% set?

Coming back to comments made by the reviewer, we want to accomplish two things: that the genomes we present can be compared and that we can provide a robust data-set for further study. As we know different methods were used (most significantly PacBio vs Nanopore). Therefore, we decided on this high stringency.

- Please change 'effector genes' to 'putative effector genes'

This has been changed.

- Supp Fig2: what is alpha?

This was an artifact of the figure assembly. This has been corrected.

- Supp Fig22: please use a more scientific layout for this plot, i.e. bar charts next to each other.

This has been changed.

- Line 94: please change 'frustrated' to 'hampered' or similar

This has been changed to "hampered".

- Line 102: these references are outdated. There are now the Pt76 and Pca203 and Pst134 genomes

Many thanks for bringing this to our attention. We have updated the references.

- Line 110: should be haplotypes instead of haplotigs

This has been changed.

REVIEWERS' COMMENTS

Reviewer #1 (Remarks to the Author):

Authors have adequately addressed my previous concerns, and the manuscript is much improved. Below are a few additional changes recommended to supplementary figures.

Supplementary Figure 4, Supplementary Figure 21 title - change "predicated" to "predicted"

Supplementary Figure 10 - What is the "0.5" at the bottom of the figure? Also, divergence times are indicated at the nodes, correct? Please state this in the legend and indicate if it is millions of years.

Supplementary Figure 21 title - change "predication" to "prediction"

Reviewer #2 (Remarks to the Author):

The authors have adequately addressed reviewer comments. Limitations and technical information have been updated. No assembly, annotation, comparative genomics project is perfect, but this study is of high quality, will be a good resource, and presents interesting new avenues for further study.

Reviewer #3 (Remarks to the Author):

Thanks to the authors for addressing my comments. Whilst I still disagree with some of the methods/findings (e.g. the polishing of the PacBio genomes with long reads only and not with high accuracy Illumina reads or the claim that Albacore performs as well as the current versions of Guppy), the revisions are overall acceptable and have improved the manuscript.

REVIEWERS' COMMENTS

Reviewer #1 (Remarks to the Author):

R: Authors have adequately addressed my previous concerns, and the manuscript is much improved. Below are a few additional changes recommended to supplementary figures.

A: We are happy to have addressed to concerns and thank the reviewer for the feedback and careful reading our manuscript.

R: Supplementary Figure 4, Supplementary Figure 21 title - change "predicated" to "predicted"

A: We have changed accordingly.

R: Supplementary Figure 10 - What is the "0.5" at the bottom of the figure? Also, divergence times are indicated at the nodes, correct? Please state this in the legend and indicate if it is millions of years.

A: We have clarified the legend accordingly now stating: The divergence times are indicated at the nodes in millions of years (My). The scale bar, 0.5 My is shown at the bottom of the phylogenetic tree.

R: Supplementary Figure 21 title – change “predication” to “prediction”

A: We have changed accordingly

Reviewer #2 (Remarks to the Author):

R: The authors have adequately addressed reviewer comments. Limitations and technical information have been updated. No assembly, annotation, comparative genomics project is perfect, but this study is of high quality, will be a good resource, and presents interesting new avenues for further study.

A: We thank the reviewer for the careful reading and feedback on our manuscript.

Reviewer #3 (Remarks to the Author):

R: Thanks to the authors for addressing my comments. Whilst I still disagree with some of the methods/findings (e.g. the polishing of the PacBio genomes with long reads only and not with high accuracy Illumina reads or the claim that Albacore performs as well as the current versions of Guppy), the revisions are overall acceptable and have improved the manuscript.

A: We thank the reviewer for the careful reading and feedback on our manuscript.